# LogART: Pushing the Limit of Efficient Logarithmic Post-Training Quantization

**Jiawei Xu**[1], **Yi Zheng**[2], **Chenghe Sun**[1], **Taiyu Zhou**[1], **Zuqi Zhang**[1], **Jie Li**[3],
**Lirong Zheng**[2], **Zhuo Zou**[2]
[1]University of Macau, [2]Fudan University, [3]South China University of Technology
jiaweixu@um.edu.mo, zhuo@fudan.edu.cn

## Abstract

Efficient deployment of deep neural networks increasingly relies on Post-Training Quantization (PTQ). Logarithmic PTQ, in particular, promises multiplier-free hardware efficiency, but its performance is often limited by the nonlinear and symmetric quantization grid and standard rounding-to-nearest (RTN) approach. While learnable rounding has significantly advanced linear PTQ, its application to the non-linear and often discrete nature of logarithmic domain remains unexplored. This paper introduces learnable Logarithmic Adaptive Rounding Techniques (Log-ART) that pioneer task-aware learnable rounding specifically for the logarithmic domain. LogART further extends the learnable rounding strategy to flexibly support outlier-aware, asymmetric, and hardware-friendly dynamic logarithmic bases, determined in a distribution-aware manner using an efficient search strategy. Extensive experiments demonstrate that LogART achieves state-of-the-art accuracy while maintaining efficiency in quantizing models across various architectures and ultra-low bitwidths, outperforming existing logarithmic PTQ methods and paving the way for more effective hardware deployment. The code is available at https://github.com/logart-lab/logart.

## 1 Introduction

Artificial intelligence (AI) has entered the era of large-scale models, with large language models (LLMs) Touvron et al. (2023); Biderman et al. (2023) reaching billions to trillions of parameters and straining existing infrastructure. Their deployment requires significant computational and memory resources, posing challenges for scalability and practical adoption. This necessitates effective model compression and efficient inference techniques, particularly for deploying these models on edge devices with tight constraints on computation, memory, and power consumption.

Quantization is a widely adopted technique for compressing deep neural networks and accelerating inference, facilitating deployment on resource-constrained edge devices. Two main quantization approaches are commonly used: Quantization-Aware Training (QAT) Liu et al. (2023); Chen et al. (2024) and Post-Training Quantization (PTQ) Nagel et al. (2020); Li et al. (2021). QAT introduces quantization during training or re-training, allowing the model to adapt to quantization noises and typically yielding higher accuracy. However, QAT requires access to the original training data and involves training phases, which is often impractical for large-scale models due to data privacy concerns, proprietary constraints, and the substantial computational resources required. In contrast, PTQ applies quantization to a pre-trained model using only a small calibration set Nagel et al. (2020); Li et al. (2021); Kim et al. (2024a); Wu et al. (2024), or in some cases no data at all Lee et al. (2017); Xu et al. (2020), offering advantages in compact model production speed and efficiency.

PTQ techniques are broadly categorized into linear Fang et al. (2020); Kwon et al. (2024); Gong et al. (2025) and non-linear Lee et al. (2017); Xu et al. (2020); Lin et al. (2022); Li et al. (2023); Wu et al. (2024) methods. Linear PTQ simplifies hardware implementation but struggles with bell-shaped distributions often observed. Logarithmic PTQ, a representative non-linear method, offers two key advantages: 1) its non-uniform levels align better with bell-shaped and long-tailed distributions, often outperforming linear PTQ at low bitwidths Lin et al. (2022); 2) base-2 logarithmic PTQ can boost hardware efficiency by replacing bulky multipliers with shifters or adders Xu et al. (2018; 2023). To

further improve performance, various optimization strategies have been proposed, including base $\sqrt{2}$ Li et al. (2023) and distribution-aware flexible bases Wu et al. (2024). However, logarithmic PTQ still faces key challenges, including performance bottlenecks at ultra-low bitwidths due to inherent symmetry and lack of outlier awareness, as well as limited adaptability to emerging models. Critically, most logarithmic PTQ methods use rounding-to-nearest (RTN), which is known to be suboptimal compared to task-aware learnable rounding Li et al. (2021); Kim et al. (2024a). Integrating it with logarithmic PTQ remains a challenge due to the non-linearity of logarithmic mapping, the non-differentiability of rounding in the logarithmic domain, and the discrete nature of mixed bases.

In this paper, we propose **Log**arithmic **A**daptive **R**ounding **T**echniques (**LogART**) for PTQ. The key idea is to enable fast learnable logarithmic rounding with outlier-resilient, asymmetric, dynamic and hardware-friendly bases, optimizing both accuracy and efficiency. Main contributions are:

- We propose LogART, a novel PTQ method that learns the optimal rounding for logarithmic quantization using a small set of unlabeled calibration data. LogART addresses both the non-learnability of logarithmic rounding and the non-differentiability in dynamic bases.

- We extend LogART with a novel quantizer that enables outlier-aware, asymmetric, multi-base, and hardware-friendly logarithmic quantization, using an efficient search strategy to rapidly identify optimal hyperparameters.

- From extensive experiments on LLMs, covolutional neural networks (CNNs), and vision transformers, we demonstrate that LogART outperforms state-of-the-art approaches in balancing accuracy, quantization runtime and hardware efficiency.

## 2    RELATED WORK

### 2.1    PTQ FOR NNs

The key goal of PTQ is to convert the weights and/or activations of a pre-trained NN model from floating-point to low-bitwidth fixed-point representations. Weights are static parameters determined after training, whereas activations are dynamic, input-dependent values generated during inference. While weight distributions remain fixed, activation distributions can vary significantly across layers and inputs. Prior studies have shown that activation PTQ can benefit from techniques such as LSQ Esser et al. (2020), whereas weight PTQ can leverage learnable rounding schemes Nagel et al. (2020); Li et al. (2021); Kim et al. (2024a). Focusing on learnable logarithmic adaptive rounding techniques for PTQ, this work specifically targets weight quantization.

PTQ techniques have been successfully applied across a variety of NN architectures. For CNNs, methods like BRECQ Li et al. (2021) leverage block-wise reconstruction to achieve high accuracy even at ultra-low bitwidths by minimizing quantization error locally through learnable weight rounding. As Transformer-based models Vaswani et al. (2017); Touvron et al. (2023); Dosovitskiy et al. (2021) have become increasingly prominent in vision and natural language processing, specialized PTQ methods have been developed to address the unique challenges posed by self-attention mechanisms Lin et al. (2022); Li et al. (2023); Wu et al. (2024; 2025). More recently, with the rise of LLMs, PTQ has become indispensable. Techniques such as GPTQ Frantar et al. (2023), aespa Kim et al. (2024a), and AWQ Lin et al. (2024) address the challenges of quantizing massive models while maintaining performance and enabling cost-effective quantization.

Learnable weight rounding schemes offer an effective way to significantly improve PTQ accuracy compared to standard RTN, especially at low bitwidths. First introduced by AdaRound Nagel et al. (2020), learnable weight rounding has pushed the limit of linear quantization into 4-bit PTQ regime. Techniques such as FlexRound Lee et al. (2023), aespa Kim et al. (2024a) and APHQ Wu et al. (2025) have further adapted and extended learnable optimization principles for linear PTQ. However, while highly effective for linear PTQ, learnable weight rounding schemes are not directly applicable to the inherently non-linear nature of logarithmic PTQ. Existing logarithmic PTQ methods have instead focused on optimizing other quantization parameters, such as adaptive bases or scaling factors, often relying on complex hyperparameter search strategies Xu et al. (2023); Wu et al. (2024).

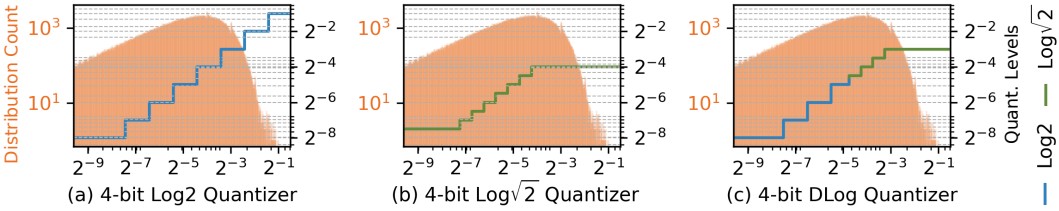

Figure 1: Illustration of (a) Log2, (b) Log$\sqrt{2}$, and (c) DLog quantization applied to a sample weight distribution (in orange) from OPT-125M model.

## 2.2 LOGARITHMIC PTQ

Logarithmic PTQ employs quantization levels that are exponentially spaced, corresponding to uniform steps in the logarithmic domain. Key variants within this family primarily differ by their chosen base, including Log2 Lee et al. (2017); Lin et al. (2022), Log$\sqrt{2}$ Xu et al. (2018); Li et al. (2023), dynamic log (DLog) Xu et al. (2020; 2023).

Log2 Quantization utilizes powers-of-two as the discrete quantization levels Lee et al. (2017):

$$Quant : \mathbf{Q_W} = \text{clamp}\left(\left\lfloor -\log_2\left(\frac{|\mathbf{W}|}{s}\right)\right\rceil, 0, 2^{N-1} - 1\right), \quad s = 2^{\lfloor \log_2(\max(|\mathbf{W}|))\rceil}, \quad (1)$$

$$Dequant : \widehat{\mathbf{W}} = s \cdot \text{sgn}(\mathbf{W}) \odot 2^{-\mathbf{Q_W}}, \quad (2)$$

where $|\cdot|$ generates the element-wise absolute value, $\log_a(\cdot)$ is the base-$a$ logarithm function, $\lfloor \cdot \rceil$ denotes the RTN function, $\text{sgn}(\cdot)$ is the sign function, $\odot$ denotes the Hadamard product, and $\text{clamp}(\mathbf{R}, a, b)$ limits the value $\mathbf{R}$ to a closed interval $[a, b]$. Here, the floating-point weight matrix $\mathbf{W}$ is quantized into an $N$-bit (sign bit included) integer-form $\mathbf{Q_W}$, from which the dequantized approximation $\widehat{\mathbf{W}}$ is derived. A key advantage of Log2 quantization is its hardware efficiency through the use of shifters instead of multipliers. However, the power-of-two scale results in coarse quantization steps especially at important large values, limiting the achievable accuracy.

Log$\sqrt{2}$ Quantization addresses this limitation by employing powers of $\sqrt{2}$ to provide finer granularity and improve the achievable accuracy ceiling Xu et al. (2018); Li et al. (2023):

$$Quant : \mathbf{Q_W} = \text{clamp}\left(\left\lfloor -\log_{\sqrt{2}}\left(\frac{|\mathbf{W}|}{s}\right)\right\rceil, 0, 2^{N-1} - 1\right), \quad s = \sqrt{2}^{\lfloor \log_{\sqrt{2}}(\max(|\mathbf{W}|))\rceil}, \quad (3)$$

$$Dequant : \widehat{\mathbf{W}} = s \cdot \text{sgn}(\mathbf{W}) \odot \sqrt{2}^{-\mathbf{Q_W}}. \quad (4)$$

A notable drawback of Log$\sqrt{2}$ is its reduced hardware-friendliness as multiplication by $\sqrt{2}$ cannot be implemented using simple shifters. Hardware acceleration strategies have been explored to compensate, such as the combined look-up table (LUT) and shifter approach Wu et al. (2024). Furthermore, it suffers from severe truncation errors that degrade accuracy, especially at low bitwidths.

DLog Quantization preserves fine-grained representation for large values while reducing the quantization gap near zero by combining base 2 and $\sqrt{2}$ (Figure 1). Optimizing the adaptive base allocation, which is critical for achieving high accuracy at low bitwidths, often relies on complex and time-consuming hyperparameter search strategies. Current logarithmic PTQ techniques face three key limitations: 1) **Inherent symmetric quantization grid** struggles with the asymmetric data distributions common in LLMs. 2) **High sensitivity to outliers**, which often leads to severe performance degradation. 3) **Reliance on the simple $\lfloor \cdot \rceil$ function**, which underperforms compared to task-aware learnable rounding methods proven effective in linear PTQ Nagel et al. (2020); Li et al. (2021); Kim et al. (2024a). Effectively integrating learnable rounding into logarithmic PTQ remains a significant challenge. Key difficulties include the non-linearity of the logarithmic mapping, the non-differentiability of rounding in the logarithmic domain, and the discrete nature of dynamic bases, all of which hinder gradient-based optimization.

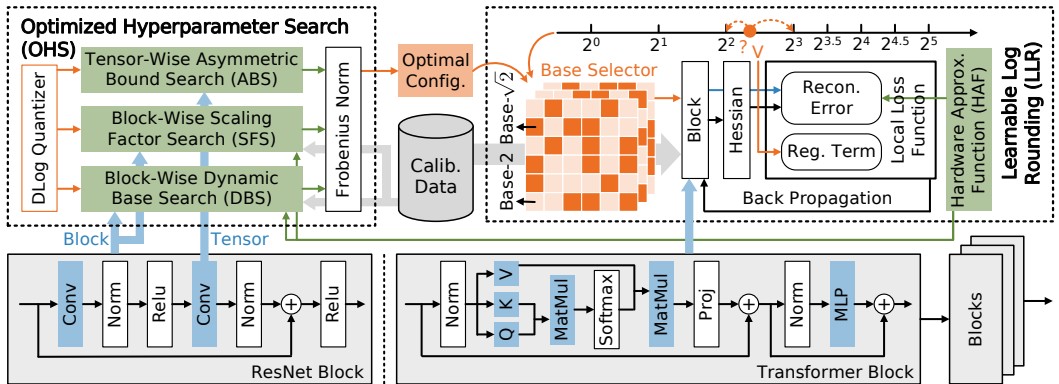

Figure 2: The overall LogART framework consists of two key components: OHS and LLR. OHS searches for optimal hyperparameter configurations in an asymmetry-aware, outlier-resilient, and multi-base manner. LLR replaces RTN with learnable element-wise rounding that minimizes local reconstruction loss while absorbing hardware approximation noise during calibration.

## 3 METHODOLOGY

LogART (Figure 2) starts by formulating an efficient learnable rounding algorithm for base-2 logarithmic PTQ to optimize the quantization-induced task loss. We then extend LogART to support outlier-aware, asymmetric, multi-base, and hardware-friendly dynamic logarithmic bases, enhancing practical performance. The searching strategy for hyperparameters and the hardware implementation of $\sqrt{2}$-based computation are also analyzed.

### 3.1 LEARNABLE LOGARITHMIC ROUNDING (LLR)

Inspired by AdaRound Nagel et al. (2020), LogART is the first to enable learnable rounding in the base-2 logarithmic domain by replacing the RTN operation in Eq. (1) with a floor operation $\lfloor \cdot \rfloor$, while a learnable variable $\mathbf{R}$ determines whether each weight is rounded down or up to get the soft-quantized $\widetilde{\mathbf{W}}$:

$$Quant : \mathbf{Q_W} = \text{clamp}\left(\left\lfloor -\log_2\left(\frac{|\mathbf{W}|}{s}\right)\right\rfloor + \sigma\left(\mathbf{R}\right), 0, 2^{N-1} - 1\right), \quad (5)$$

$$Dequant : \widetilde{\mathbf{W}} = s \cdot \text{sgn}(\mathbf{W}) \odot 2^{-\mathbf{Q_W}}. \quad (6)$$

The variable $\mathbf{R}$ are optimized by minimizing a task-aware reconstruction error with a regularization term that encourages the sigmoid-like function $\sigma\left(\mathbf{R}\right)$ towards either 0 or 1:

$$\arg\min_{\mathbf{R}} \mathbb{E}\left[\mathcal{L}\left(\mathbf{\Delta W}\right)\right] + \lambda \sum_{i,j}\left(1 - |2\sigma(\mathbf{R}_{ij}) - 1|^{\beta}\right), \quad (7)$$

here, $\mathcal{L}\left(\cdot\right)$ denotes the task loss function, which can be flexibly configured based on the reconstruction granularity, such as layer-wise reconstruction Frantar et al. (2023):

$$\mathbb{E}\left[\mathcal{L}\left(\mathbf{\Delta W}\right)\right] = \mathbb{E}\left[\|\mathbf{\Delta W X}\|_F^2\right] = \text{tr}(\mathbf{\Delta W} \cdot \mathbb{E}\left[\mathbf{X X}^\top\right] \cdot \mathbf{\Delta W}^\top), \quad \mathbf{\Delta W} = \mathbf{W} - \widetilde{\mathbf{W}}, \quad (8)$$

where $\|\cdot\|_F^2$ denotes the Frobenius norm, and $\text{tr}(\cdot)$ denotes the trace of a square matrix. Block-wise reconstruction is also supported, and the gradient of LLR is analyzed and compared with linear learnable rounding in Appendix A.

### 3.2 EXTENDING LOGART FOR NOVEL LOGARITHMIC QUANTIZER

To overcome the limitations of the fixed Log2 quantization, LogART is extended to support novel multi-base, outlier-resilient, and asymmetric logarithmic quantizer.

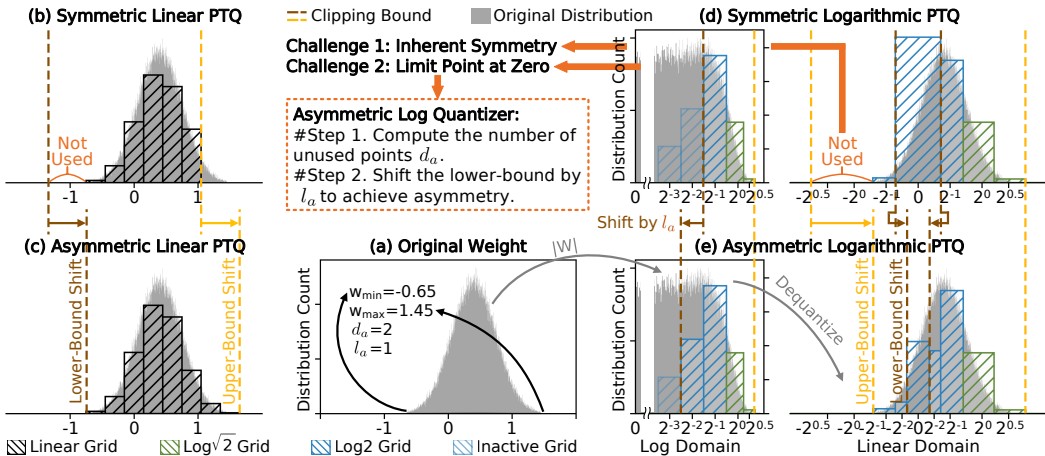

Figure 3: Illustration of the asymmetry challenge in logarithmic quantization and the proposed solution. (a) Original weight distribution. In linear PTQ, switching from symmetry (b) to asymmetry (c) is easily achieved by using a zero-point to shift the clamping bounds. (d) Conventional logarithmic PTQ is inherently symmetric. (e) The proposed asymmetric logarithmic quantizer.

**Dynamic Base Quantizer** uses base $\sqrt{2}$ for large values and base 2 to smaller values, with the ratio between them adapted in a distribution-aware manner. Given a selected base configuration, the resulting $N$-bit integer-form quantization codebook contains $n_1$ base-$\sqrt{2}$ codes and $n_2$ base-2 codes:

$$n_1 + n_2 = 2^{N-1} - 1, \quad n_1, n_2 \in \mathbb{Z} \cap [0, 2^{N-1} - 1]. \tag{9}$$

The threshold $t$ is then calculated to separate the input values into regions where either base-2 or base-$\sqrt{2}$ quantization is applied, from which the element-wise scaling factor $\mathbf{S}$, upper bound $\mathbf{U}$, and base selector $\mathbf{B}$ are constructed:

$$t = \sqrt{2}^{\frac{m-n_1+1}{2} + \left\lfloor \frac{m-n_1}{2} \right\rfloor}, \quad m = \left\lfloor \log_{\sqrt{2}}\left(\max(|\mathbf{W}|)\right) \right\rceil, \tag{10}$$

$$\mathbf{B}_{ij} = \begin{cases} \sqrt{2}, & |\mathbf{W}_{ij}| \geq t \\ 2, & |\mathbf{W}_{ij}| < t \end{cases}, \quad \mathbf{S}_{ij} = \begin{cases} \sqrt{2}^m, & |\mathbf{W}_{ij}| \geq t \\ 2^{\left\lfloor \frac{m-n_1}{2} \right\rfloor}, & |\mathbf{W}_{ij}| < t \end{cases}, \quad \mathbf{U}_{ij} = \begin{cases} n_1 - 1, & |\mathbf{W}_{ij}| \geq t \\ n_2 - 1, & |\mathbf{W}_{ij}| < t \end{cases}. \tag{11}$$

Dynamic quantization applies an element-wise conversion to the logarithmic domain:

$$Quant : \mathbf{Q_W} = \text{clamp}\left(\left\lfloor -\log_{\mathbf{B}}\left(\frac{|\mathbf{W}|}{\mathbf{S}}\right) \right\rceil + \sigma\left(\mathbf{R}\right), 0, \mathbf{U}\right), \tag{12}$$

$$Dequant : \widetilde{\mathbf{W}} = \mathbf{S} \cdot \text{sgn}(\mathbf{W}) \odot \mathbf{B}^{-\mathbf{Q_W}}. \tag{13}$$

**Asymmetric Quantizer** addresses the inherent symmetry of existing logarithmic quantization with a simple yet effective adaptive bound $l_a$. As shown in Figure 3, most logarithmic quantization methods begin by taking the absolute value of the weight tensor and then applying logarithmic quantization symmetrically, which fails to capture the naturally asymmetric weight distributions, particularly prominent in LLMs. Furthermore, in contrast to linear quantization, logarithmic quantization cannot rely on a simple zero-point to shift bounds for asymmetry due to its non-uniform spacing near zero. To overcome these challenges, LogART introduces the first asymmetric logarithmic quantizer that allocates different numbers of codes to positive and negative weights, based on the maximum $w_{\max}$ and minimum $w_{\min}$ weight values:

$$w_h = \max(w_{\max}, -w_{\min}), \quad w_l = \min(w_{\max}, -w_{\min}), \quad l_a = \lfloor d_a/2 \rfloor, \tag{14}$$

$$d_a = \begin{cases} \left\lfloor \log_{\sqrt{2}}(w_h) \right\rceil - \left\lfloor \log_{\sqrt{2}}(w_l) \right\rceil, & w_l \geq t \\ n_1 + \left\lfloor \frac{m-n_1}{2} \right\rfloor - \left\lfloor \log_2(w_l) \right\rceil, & w_l < t \end{cases}, \quad \mathbf{U}_{ij} = \begin{cases} n_1 - 1, & |\mathbf{W}_{ij}| \geq t \\ n_2 - 1 + l_a, & |\mathbf{W}_{ij}| < t \end{cases}, \tag{15}$$

$$Quant : \mathbf{Q_W} = \text{clamp}\left(\left\lfloor -\log_{\mathbf{B}}\left(\frac{|\mathbf{W}|}{\mathbf{S}}\right) \right\rceil + \sigma\left(\mathbf{R}\right), 0, \mathbf{U}\right). \tag{16}$$

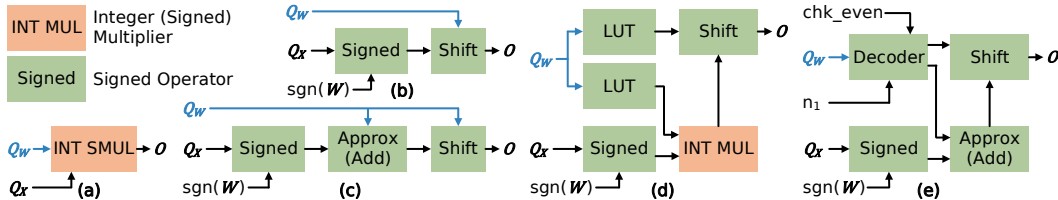

Figure 4: Flow charts of AE designs for: (a) linear quantization, (b) base-2 logarithmic quantization, (c) base-$\sqrt{2}$ quantization, (d) AdaLog quantization, and (e) LogART quantization.

**Outlier-Resilient Quantizer** addresses the strong outlier sensitivity of logarithmic PTQ by introducing a searchable hyperparameter $s_{\text{of}}$. Instead of determining the quantization range from the absolute maximum value, which is easily skewed by outliers, $s_{\text{of}}$ enables adaptive clipping of extreme values:

$$Quant : \mathbf{Q_W} = \text{clamp}\left(\left\lfloor -\log_{\mathbf{B}}\left(\frac{|\mathbf{W}|}{s_{\text{of}} \cdot \mathbf{S}}\right)\right\rceil + \sigma\left(\mathbf{R}\right), l_a, \mathbf{U}\right). \tag{17}$$

### 3.3 OPTIMIZED HYPERPARAMETER SEARCH (OHS)

While element-wise LLR helps reduce local quantization error, the overall performance of PTQ critically depends on the choice of quantizer hyperparameters. This motivates the exploration of an efficient search strategy to identify optimal hyperparameters. Our proposed OHS solution is a multi-level search strategy composed of three components:

- **Tensor-Wise Asymmetric Bound Search (ABS)**: A calibration-free step that determines the per-channel $l_a$ based solely on the minimum and maximum weight values, as previously defined in Eq. (14-16), making it both simple and efficient.
- **Block-Wise Scaling Factor Search (SFS)**: Searches for the optimal per-channel scaling factor $s_{\text{of}}$ by minimizing the reconstruction error at the block level (e.g., a residual bottleneck in ResNet or an attention module in a Transformer).
- **Block-Wise Dynamic Base Search (DBS)**: Adaptively assigns different $n_1{:}n_2$ logarithmic base configurations across weight channels, guided by block-level reconstruction error.

ABS requires no calibration, while SFS and DBS are jointly optimized using a small calibration set by minimizing the Frobenius norm of the block-wise reconstruction error:

$$\arg\min_{s_{\text{of}},n_1,n_2} \mathbb{E}\left[\|\mathcal{L}(\mathbf{\Delta W}, \mathbf{X})\|_F^2\right]. \tag{18}$$

The granularity of OHS can vary from tensor-wise to block-wise, to align with different reconstruction levels. We analyze the impact of these search granularities, both with and without LLR, in Appendix B. Our findings reveal a strong synergy between the multi-level OHS and LLR, leading to higher quantized model accuracy and significantly faster convergence.

### 3.4 HARDWARE APPROXIMATION FUNCTION (HAF)

Beyond model accuracy, hardware efficiency is a critical consideration for the practical deployment of quantized models. LogART, particularly when employing dynamic logarithmic bases, necessitates a careful analysis of the associated hardware implementation complexity and efficiency.

To ensure a hardware-friendly design for the Arithmetic Element (AE) that handles computations involving $\sqrt{2}$ in LogART, we introduce a HAF module. HAF is incorporated into the quantized forward pass during the OHS and LLR process, allowing the induced error to be absorbed as noise during the learning process. The core principle of HAF is to replace multiplications involving $\sqrt{2}$ with simple shift-add operations through the hardware approximation:

$$\sqrt{2} \approx \text{SDE}(\sqrt{2}, K) = \sum_{k=1}^{K} a_k \cdot \frac{1}{2^{d_k}}, \quad \text{where } a_k \in \{-1, +1\}, \ d_k \in \mathbb{N}, \ d_1 < d_2 < \cdots . \tag{19}$$

Table 1: Ablation results of LogART key components on LLMs with 3-bit per-channel weight quantization, evaluated in terms of calibration data (from WikiText-2) dependency, PPL on WikiText-2 dataset, time cost, and GPU memory cost.

| DBS | SFS | ABS | LLR | Calib. Data | OPT-125M | | | LLaMA2-7B | | |
|---|---|---|---|---|---|---|---|---|---|---|
| | | | | | PPL | Time | Memory | PPL | Time | Memory |
| × | × | × | × | - | 170.64 | 0.7 s | 0.40 GB | 60.16 | 13.0 s | 9.8 GB |
| ✓ | × | × | × | 32 | 66.63 | 3.8 s | 0.75 GB | 18.49 | 83.2 s | 20.9 GB |
| ✓ | ✓ | × | × | 32 | 36.10 | 16.8 s | 0.75 GB | 6.56 | 17.9 min | 20.9 GB |
| ✓ | ✓ | ✓ | × | 32 | 34.29 | 17.0 s | 0.75 GB | 6.45 | 17.9 min | 20.9 GB |
| ✓ | ✓ | ✓ | ✓ | 32 | 31.15 | 75.1 s | 0.75 GB | 6.14 | 74.2 min | 20.9 GB |

Table 2: Ablation results of LogART key components on CNN and vision transformer models with 4-bit per-channel weight quantization, evaluated in terms of top-1 accuracy and GPU runtime.

| DBS | SFS | ABS | LLR | ResNet18 | | MobileNetV2 | | ViT-Base | | DeiT-Tiny | |
|---|---|---|---|---|---|---|---|---|---|---|---|
| | | | | Acc(%) | Time | Acc(%) | Time | Acc(%) | Time | Acc(%) | Time |
| × | × | × | × | 31.53 | 0.9 s | 1.22 | 1.9 s | 79.55 | 0.0 min | 57.25 | 0.0 min |
| ✓ | × | × | × | 68.45 | 1.0 s | 66.91 | 2.3 s | 84.24 | 0.1 min | 69.17 | 0.0 min |
| ✓ | ✓ | × | × | 69.69 | 10.8 s | 69.47 | 22.0 s | 84.61 | 2.8 min | 70.29 | 1.4 min |
| ✓ | ✓ | ✓ | × | 69.89 | 11.6 s | 69.86 | 24.4 s | 84.67 | 2.8 min | 70.40 | 1.4 min |
| ✓ | ✓ | ✓ | ✓ | 70.79 | 72.3 s | 71.62 | 156.6 s | 85.02 | 10.9 min | 71.62 | 5.6 min |

Here, $\text{SDE}(\sqrt{2}, K)$ denotes the $K$-term Signed Dyadic Expansion of the real number $\sqrt{2}$. For example, a 2-term ($K = 2$) approximation is $\sqrt{2} \approx 2^0 + 2^{-1}$.

In Appendix C, we provide a detailed analysis of the hardware complexity of the AE designs in Figure 4 for different PTQ methods. Our evaluation demonstrates that the multiplier-free LogART AE consistently achieves a favorable balance between accuracy and hardware efficiency.

## 4 EXPERIMENT

To evaluate the effectiveness of LogART, we conduct experiments across various models. An ablation study is performed to analyze the contribution of key components. LogART is benchmarked against state-of-the-art (SOTA) PTQ methods in terms of accuracy, GPU memory, runtime, data dependency, and AE hardware efficiency. Additional experiments and results are provided in the Appendix.

### 4.1 EXPERIMENTAL SETUP

**Datasets and Models.** We quantize publicly available pre-trained full-precision models, including LLMs: OPT Zhang et al. (2022) and LLaMA Touvron et al. (2023) series on the WikiText-2 Merity et al. (2017) and C4 Raffel et al. (2020) dataset; CNNs: ResNet He et al. (2016) and MobileNetV2 Sandler et al. (2018) on the ImageNet Russakovsky et al. (2015) dataset; and vision transformers: ViT Dosovitskiy et al. (2021) and DeiT Touvron et al. (2021) on ImageNet.

**Quantization Details.** Our evaluation focuses on 3-bit and 4-bit weight-only quantization, a common setting that significantly accelerates large models by reducing memory movement overhead Kim et al. (2024b). While our experiments focus on weights, LogART is fully compatible with various existing activation quantization methods, such as SmoothQuant Xiao et al. (2023), AdaLog Wu et al. (2024), QuaRot Ashkboos et al. (2024), ERQ Zhong et al. (2024), and APHQ Wu et al. (2025). For the calibration dataset, we randomly sample 32 segments of 2048 tokens each from the WikiText-2 or C4 dataset for language tasks, and 2048 unlabeled images from ImageNet for vision tasks. The primary evaluation metrics are perplexity (PPL) for language modeling and top-1 accuracy for image classification. All experiments are conducted on a single NVIDIA RTX 5090D GPU (32 GB).

Table 3: Performance (PPL), GPU runtime, and memory usage of 3-bit weight quantization of LogART and existing PTQ methods on LLM models. (Calibration data from C4)

| Model | Method | FP16 | GPTQ | BRECQ | AffineQuant | aespa | LogART |
|-------|--------|------|------|-------|-------------|-------|--------|
| | Domain | - | Linear | Linear | Linear | Linear | Log |
| OPT-125M | PPL (WikiText-2) | 27.65 | 52.95 | 34.07 | 36.15 | 34.53 | **31.52** |
| | PPL (C4) | 26.56 | 42.88 | 31.44 | 32.78 | 31.41 | **29.98** |
| | Runtime | - | 19.8 s | 1.06 hr | 16.7 min | 2.81 min | 1.25 min |
| | Memory (GB) | - | 1.05 | 3.07 | 3.14 | 1.52 | 0.75 |
| OPT-1.3B | PPL (WikiText-2) | 14.63 | 20.36 | 16.09 | 17.26 | 16.07 | **15.53** |
| | PPL (C4) | 16.07 | 20.53 | 17.46 | 18.27 | 17.40 | **17.29** |
| | Runtime | - | 2.44 min | 4.57 hr | 1.10 hr | 31.8 min | 11.0 min |
| | Memory (GB) | - | 4.08 | 15.4 | 9.23 | 5.07 | 4.18 |
| OPT-6.7B | PPL (WikiText-2) | 10.86 | 13.01 | OOM* | 12.30 | 11.35 | **11.11** |
| | PPL (C4) | 12.71 | 14.61 | OOM* | 13.80 | 13.42 | **13.37** |
| | Runtime | - | 12.2 min | OOM* | 4.54 hr | 4.42 hr | 1.45 hr |
| | Memory (GB) | - | 12.3 | OOM* | 23.4 | 14.3 | 17.1 |
| LLaMA2-7B | PPL (WikiText-2) | 5.47 | 8.66 | OOM* | 6.80 | 6.45 | **6.31** |
| | PPL (C4) | 7.26 | 11.24 | OOM* | 9.06 | 8.51 | **8.38** |
| | Runtime | - | 10.4 min | OOM* | 5.01 hr | 3.39 hr | 1.24 hr |
| | Memory (GB) | - | 8.28 | OOM* | 23.7 | 21.2 | 20.8 |
| LLaMA3-8B | PPL (WikiText-2) | 6.14 | 11.14 | OOM* | - | 8.95 | **8.19** |
| | PPL (C4) | 9.45 | 13.86 | OOM* | - | 12.59 | **12.44** |
| | Runtime | - | 12.5 min | OOM* | - | 3.68 hr | 1.46 hr |
| | Memory (GB) | - | 10.6 | OOM* | - | 20.0 | 21.6 |

\* OOM indicates that an out-of-memory (OOM) error occurred during quantization.

## 4.2 COMPARISON WITH SOTA APPROACHES

**Ablation Study.** Our ablation study, detailed in Table 1, Table 2 and Appendix D, validates the distinct and complementary contributions of each LogART component: DBS, SFS, ABS and LLR. The results identify SFS and LLR as the most impactful modules. SFS delivers a significant reduction in perplexity with a moderate time overhead. Specifically, it lowers PPL on LLaMA2-7B from 9.74 to 6.24 when added to the LLR baseline at the cost of extra 4.5 minutes. The benefits of LLR are demonstrated by the significant accuracy boost it provides over the RTN counterpart. The advantage of DBS is clearly evident when applied alone. Compared to a fixed Log2 base, DBS reduces PPL by more than half, owing to its enhanced flexibility in capturing diverse weight characteristics. ABS is highly efficient, providing consistent accuracy gains without calibration data and with negligible overhead. Its impact depends on model weight distributions, with larger improvements observed on LLaMA2-7B than on OPT-125M. Ultimately, the study reveals a powerful synergistic effect, as the full LogART with all components enabled consistently achieves the best performance. This confirms that each module is a vital and complementary contributor to the final result.

**Comparison on LLMs.** We benchmark the proposed LogART against several SOTA PTQ methods on LLMs, with per-channel weight quantization results presented in Table 3. For our method, we enable the full OHS and LLR, performing optimization for 500 iterations using the Adam optimizer. We employ a CosineAnnealingLR scheduler with a learning rate decaying from 0.05 to 0.015 and a rounding loss weight of 1. We compare against strong baselines: the backpropagation-free GPTQ Frantar et al. (2023), the classic optimization-based BRECQ Li et al. (2021), and its efficient successors AffineQuant Ma et al. (2024) and aespa Kim et al. (2024a). For BRECQ, we employ the hyperparameter settings provided in Li et al. (2021). For AffineQuant and aespa, we use their official implementations[1] and reported settings. For a fair comparison, we use the calibration dataset from C4, and additional results using the WikiText-2 dataset are available in Appendix E.

LogART is the first logarithmic PTQ method to scale effectively to LLMs at 3-bit, and it consistently achieves SOTA accuracy across all tested LLMs. While GPTQ is the fastest method due to its RTN-

[1]https://github.com/bytedance/AffineQuant, https://github.com/SamsungLabs/aespa

Table 4: Comparison of top-1 accuracy on ImageNet and GPU runtime (in minutes) for different per-channel 4-bit weight PTQ methods on CNN models.

| Model | Method | FP16 | AdaRound | BRECQ | FlexRound | LogNet | SLogII | LogART |
|-------|--------|------|----------|-------|-----------|--------|--------|--------|
|       | Domain | -    | Linear   | Linear | Linear   | Log    | Log    | Log    |
| ResNet18 | Acc (%) | 71.00 | 70.18 | 70.47 | 70.28 | 31.53 | 67.52 | **70.79** |
|          | Runtime | -     | 8.0   | 9.3   | -*    | 0.0   | 0.4   | 1.2   |
| ResNet50 | Acc (%) | 76.63 | 75.86 | 76.43 | 75.95 | 42.76 | 74.14 | **76.57** |
|          | Runtime | -     | 19.4  | 22.7  | -*    | 0.0   | 2.1   | 3.1   |
| MobileNetV2 | Acc (%) | 72.62 | 69.46 | 71.52 | 70.82 | 1.22  | 31.20 | **71.62** |
|             | Runtime | -     | 17.6  | 19.5  | -*    | 0.0   | 1.3   | 2.6   |

* The GPU runtime is not reported and cannot be measured, as its code is not publicly available.

Table 5: Comparison of top-1 accuracy on ImageNet and GPU runtime (in minutes) for different per-channel 4-bit weight PTQ methods on vision transformer models.

| Model | Method | FP16 | BRECQ | APHQ | AdaLog | LogNet | SLogII | LogART |
|-------|--------|------|-------|------|--------|--------|--------|--------|
|       | Domain | -    | Linear | Linear | Linear | Log   | Log    | Log    |
| ViT-Small | Acc (%) | 81.39 | 80.52 | 80.72 | 80.65 | 70.14 | 78.54 | **81.06** |
|           | Runtime | -     | 35.6  | 32.1  | 32.2  | 0.0   | 3.1   | 6.7   |
| ViT-Base  | Acc (%) | 85.10 | 84.83 | 84.79 | 84.77 | 79.55 | 83.54 | **85.02** |
|           | Runtime | -     | 91.8  | 88.9  | 91.6  | 0.0   | 6.2   | 10.9  |
| DeiT-Tiny | Acc (%) | 72.16 | 71.34 | 71.46 | 71.14 | 57.25 | 68.36 | **71.62** |
|           | Runtime | -     | 21.9  | 22.2  | 22.1  | 0.0   | 1.6   | 5.6   |
| DeiT-Base | Acc (%) | 81.98 | 81.77 | 81.71 | 81.85 | 79.53 | 81.01 | **81.92** |
|           | Runtime | -     | 92.5  | 89.0  | 91.7  | 0.0   | 6.2   | 10.9  |

based, learning-free design, LogART surpasses it with a significant accuracy advantage. Compared to optimization-based linear PTQ methods like BRECQ, AffineQuant, and aespa, LogART not only achieves lower perplexity but is also substantially faster, reducing runtime by over $24.9\times$, $3.1\times$ and $2.2\times$, respectively, with similar or lower GPU memory usage. The outstanding performance of LogART is attributed to the inherent advantages of logarithmic quantization, which better captures the data distribution in LLMs, and the synergistic effect of the proposed OHS and LLR techniques.

**Comparison on CNNs and Vision Transformers.** We compare LogART with SOTA PTQ methods on CNNs and vision transformers, with per-channel weight quantization results reported in Table 4, Table 5 and Appendix F. Baselines include linear-domain weight-rounding methods (AdaRound, BRECQ, FlexRound, APHQ, AdaLog) and logarithmic schemes (LogNet, SLogII). Although AdaLog applies logarithmic quantization to activations and linear quantization to weights, we categorize it under the linear domain in this comparison, as our primary focus is weight quantization. For BRECQ Li et al. (2021) and SLogII Xu et al. (2023), we use the official implementations with their reported settings. AdaRound Nagel et al. (2020) is re-implemented following BRECQ, and LogNet Lee et al. (2017) is re-implemented following SLogII. Results for FlexRound Lee et al. (2023) are directly quoted from the original publication. For vision transformer experiments, we implement APHQ and AdaLog with official code[2]. Since the original BRECQ does not support vision transformers, we re-implement it within the APHQ framework. For LogART, we use the same settings as in LLMs but increase LLR iterations to 2000 to ensure convergence on these architectures.

LogART consistently achieves SOTA accuracy across all evaluated CNN and vision transformer models. It significantly outperforms the logarithmic PTQ baselines, with particularly notable improvements on MobileNetV2, where it achieves >40% higher top-1 accuracy. These gains stem from the proposed OHS and LLR techniques, whereas prior logarithmic PTQ methods are limited by symmetric quantizers, tensor-wise hyperparameter search, and naive RTN policies. Compared to advanced linear PTQ methods with learnable rounding, LogART remains competitive in accuracy

---

[2]https://github.com/GoatWu/APHQ-ViT, https://github.com/GoatWu/AdaLog

Table 6: HAF evaluation and AE comparison in terms of top-1 accuracy and hardware efficiency.

| Method | W/A Domain | Hardware Evaluation | | Accuracy (%) Evaluation | | | |
|---|---|---|---|---|---|---|---|
| | | Area($\mu m^2$) | Power($\mu W$) | ViT-S | ViT-B | DeiT-T | DeiT-B |
| BRECQ | Linear/Linear | 95.8 | 6.28 | 80.47 | 84.69 | 71.05 | 81.36 |
| AdaLog | Linear/Log | 76.2 | 5.56 | 77.37 | 83.15 | 67.18 | 80.14 |
| LogART w/o HAF | Log/Linear | **53.2** | **3.45** | 80.12 | 84.36 | 70.14 | 81.51 |
| LogART w/ HAF | | | | **81.02** | **84.99** | **71.59** | **81.88** |

while offering substantially higher efficiency, achieving speedups of over 6.3×, 3.9×, 4.0×, and 3.9× against AdaRound, BRECQ, APHQ, and AdaLog, respectively. The acceleration is primarily driven by the faster convergence enabled through the synergy of OHS and LLR.

**HAF and AE Hardware Efficiency Evaluation.** The proposed HAF is highly effective, incurring minimal accuracy degradation (<0.2% on vision models and <0.2 PPL on LLMs) compared to ideal LogART, while significantly outperforming a naive hardware approximation (e.g., +8.81% accuracy on MobileNetV2). See Appendix C for experimental results. For AE hardware efficiency, we simulate LogART AE in Figure 4 under a 4-bit weight and 8-bit activation setting, with results presented in Table 6 and Appendix G. Here, activation quantization is performed in the linear domain. Power and area consumption are synthesized in a 28-nm UMC process using Synopsys Design Compiler, operating at 250 MHz and 0.9 V. BRECQ applies asymmetric linear quantization to weights and activations, requiring 8/8-bit multipliers and consuming 80.08% more area and 82.03% more power than LogART. Since AdaLog uses linear quantization for weights and logarithmic quantization for activations, we adopt 8-bit weights and 4-bit activations for a fair comparison. Compared to the LUT/multiplier/shifter-based AdaLog AE, LogART AE achieves a 43.23% area and 61.16% power advantage, confirming its superiority in both accuracy and hardware efficiency.

**Computational Overhead in Practical Deployment.** LogART achieves a dual advantage in computational overhead: low one-off offline model production cost and reduced recurring inference hardware cost. The one-off cost is incurred only during the model production phase, not during practical inference after deployment. The reported GPU runtime corresponds exactly to this production time and explicitly includes the learning iterations. For practical usage, the learned rounding parameters are frozen into the model weights. Therefore, the recurring inference cost is determined by the AE. As discussed above, the LogART AE achieves significant hardware savings. From a long-term perspective, the extra learning time is a one-off setup cost amortized over millions of inference runs, resulting in a negligible marginal cost. Overall, LogART achieves a competitive balance, offering both fast offline quantized model production and superior online inference hardware efficiency.

## 5 CONCLUSION

We propose LogART, the first PTQ scheme that integrates learnable rounding into the logarithmic domain. By combining this with the novel multi-base, outlier-resilient, and asymmetric quantizer through an efficient search strategy, LogART pushes the limit of logarithmic weight PTQ to ultra-low bitwidths on LLMs, CNNs, and vision transformers. Furthermore, LogART achieves a superior trade-off between accuracy and hardware efficiency by incorporating a practical hardware approximation directly into optimization. Extensive experiments validate that LogART not only consistently reaches SOTA accuracy but also achieves over 40% reduction in AE area and power consumption. By providing a robust and hardware-aware solution, LogART paves the way for the efficient deployment of large models on resource-constrained hardware. Future work will extend LogART to joint weight-activation quantization and explore its integration with other compression techniques.

### ACKNOWLEDGMENTS

This work was supported in part by the Science and Technology Development Fund, Macau SAR (FDCT) under Grant 0001/2025/NRP; in part by the National Natural Science Foundation of China under Grant 62476062; in part by the University of Macau under Grant MYRG-GRG2025-00283-IME; in part by the Shanghai Platform for Neuromorphic and AI Chip (NeuHelium).

REPRODUCIBILITY STATEMENT

To ensure reproducibility, we provide the full source code for LogART along with all experiment scripts in the anonymous repository linked in the abstract. All datasets used (ImageNet, C4, WikiText-2) are publicly available, and our data processing and calibration procedures are described in Section 4 and Appendix E. The experimental setup and hyperparameter configurations for all methods are detailed in Section 4. Comprehensive results, including full ablation tables and hardware analysis, are provided in Appendix B, C, D, E, F, and G to support our claims.

ETHICS STATEMENT

This work does not involve human subjects, personally identifiable information, or sensitive user data. All datasets used in our experiments (ImageNet, C4, and WikiText-2) are publicly available benchmark datasets, and we followed the standard licensing and usage guidelines associated with each. Our methods focus on post-training quantization and hardware-efficient deployment of large models such as LLMs and vision transformers. While model compression and efficiency improvements may enable broader deployment of these models, we acknowledge potential concerns about misuse, including the generation of harmful content or biased outputs inherited from pretrained models. We emphasize that our contributions are methodological and hardware-focused, and do not alter or exacerbate underlying dataset or model biases. We have taken care to report results transparently, and comply with research integrity and ethical guidelines as outlined in the ICLR Code of Ethics. The authors declare no competing interests.

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

## A GRADIENT OF TASK-AWARE LLR LOSS

The total loss function for LLR combines a task-aware reconstruction error with a regularization term. For example, under layer-wise reconstruction granularity, the loss is defined according to Eq. (8) as:

$$\mathcal{L} = \mathcal{L}_{recon} + \mathcal{L}_{reg}, \quad \mathcal{L}_{recon} = \mathbb{E}\left[\|\Delta\mathbf{W}\mathbf{X}\|_F^2\right], \quad \mathcal{L}_{reg} = \lambda \cdot f_{reg}(\mathbf{R}), \tag{20}$$

$$f_{reg}(\mathbf{R}) = \sum_{i,j}\left(1 - |2\sigma(\mathbf{R}_{ij}) - 1|^\beta\right), \tag{21}$$

For block-wise reconstruction granularity, the reconstruction loss for attention module is defined as:

$$\mathcal{L}_{recon} = \mathbb{E}\left[\left\|\mathrm{SA}(\widetilde{\mathbf{Q}}, \widetilde{\mathbf{K}}, \widetilde{\mathbf{V}}) - \mathrm{SA}(\mathbf{Q}, \mathbf{K}, \mathbf{V})\right\|_F^2\right]. \tag{22}$$

Here $\mathrm{SA}(\mathbf{Q}, \mathbf{K}, \mathbf{V})$ generates the output of the attention module, and $\Delta\mathbf{W}_{\mathbf{Q}}, \Delta\mathbf{W}_{\mathbf{K}}, \Delta\mathbf{W}_{\mathbf{V}}$ denote the quantization error of query, key, and value projections respectively. To reduce computational overhead, $\mathcal{L}_{recon}$ is further optimized in a divide-and-conquer manner Kim et al. (2024a):

$$\begin{aligned}\mathcal{L}_{recon}^Q &= \mathbb{E}\left[\left\|\mathrm{SA}(\widetilde{\mathbf{Q}}, \mathbf{K}, \mathbf{V}) - \mathrm{SA}(\mathbf{Q}, \mathbf{K}, \mathbf{V})\right\|_F^2\right] \approx \mathbb{E}\left[\|\mathbf{K}\Delta\mathbf{W}_{\mathbf{Q}}\mathbf{X}\|_F^2\right] \\ &= \mathrm{tr}\left(\mathbb{E}\left[\mathbf{K}^\top\mathbf{K}\right] \cdot \Delta\mathbf{W}_{\mathbf{Q}} \cdot \mathbb{E}\left[\mathbf{X}\mathbf{X}^\top\right] \cdot \Delta\mathbf{W}_{\mathbf{Q}}^\top\right),\end{aligned} \tag{23}$$

$$\begin{aligned}\mathcal{L}_{recon}^K &= \mathbb{E}\left[\left\|\mathrm{SA}(\mathbf{Q}, \widetilde{\mathbf{K}}, \mathbf{V}) - \mathrm{SA}(\mathbf{Q}, \mathbf{K}, \mathbf{V})\right\|_F^2\right] \approx \mathbb{E}\left[\|\mathbf{Q}\Delta\mathbf{W}_{\mathbf{K}}\mathbf{X}\|_F^2\right] \\ &= \mathrm{tr}\left(\mathbb{E}\left[\mathbf{Q}^\top\mathbf{Q}\right] \cdot \Delta\mathbf{W}_{\mathbf{K}} \cdot \mathbb{E}\left[\mathbf{X}\mathbf{X}^\top\right] \cdot \Delta\mathbf{W}_{\mathbf{K}}^\top\right),\end{aligned} \tag{24}$$

$$\begin{aligned}\mathcal{L}_{recon}^V &= \mathbb{E}\left[\left\|\mathrm{SA}(\mathbf{Q}, \mathbf{K}, \widetilde{\mathbf{V}}) - \mathrm{SA}(\mathbf{Q}, \mathbf{K}, \mathbf{V})\right\|_F^2\right] = \mathbb{E}\left[\|\Delta\mathbf{W}_{\mathbf{V}}\mathbf{X}\mathbf{A}^\top\|_F^2\right] \\ &= \mathrm{tr}\left(\Delta\mathbf{W}_{\mathbf{V}} \cdot \mathbb{E}\left[\mathbf{X}\mathbf{A}^\top\mathbf{A}\mathbf{X}^\top\right] \cdot \Delta\mathbf{W}_{\mathbf{V}}^\top\right).\end{aligned} \tag{25}$$

The aim of the LLR process is to minimize the loss function $\mathcal{L}$ by optimizing the learnable variable $\mathbf{R}$ using stochastic gradient descent. The gradient of $\mathcal{L}$ with respect to $\mathbf{R}$ is derived through the following steps. The gradient of the layer-wise reconstruction error term is computed by propagating derivatives through the quantization chain. First, the derivative of the loss $\mathcal{L}$ with respect to $\widetilde{\mathbf{W}}$ is:

$$\frac{\partial\mathcal{L}}{\partial\widetilde{\mathbf{W}}} = -2 \cdot (\mathbf{W}\mathbf{X} - \widetilde{\mathbf{W}}\mathbf{X}) \cdot \mathbf{X}^\top, \tag{26}$$

which captures the sensitivity of the Frobenius norm to deviations between the original and quantized weights. The quantized weights $\widetilde{\mathbf{W}}$ depend on the integer form $\mathbf{Q}_{\mathbf{W}}$ during logarithmic quantization as shown in Eq. (6), leading to:

$$\frac{\partial\widetilde{\mathbf{W}}}{\partial\mathbf{Q}_{\mathbf{W}}} = -s\ln 2 \cdot \mathrm{sgn}(\mathbf{W}) \odot 2^{-\mathbf{Q}_{\mathbf{W}}}. \tag{27}$$

The $\mathbf{Q}_{\mathbf{W}}$ are clamped within $[0, 2^{N-1} - 1]$ to enforce bit-width constraints, introducing a binary mask $\mathbf{M}_c$:

$$\mathbf{M}_c = \mathbb{I}(\mathbf{Q}_{\mathbf{W}} \in [0, 2^{N-1} - 1]), \tag{28}$$

that truncates gradients outside the valid range. Combined with the derivative of the sigmoid function $\sigma(\mathbf{R})$, the reconstruction gradient becomes:

$$\frac{\partial\mathcal{L}_{recon}}{\partial\mathbf{R}} = 2s\ln 2 \cdot \mathbf{M}_c \odot 2^{-\mathbf{Q}_{\mathbf{W}}} \odot \mathrm{sgn}(\mathbf{W}) \odot \left[(\mathbf{W}\mathbf{X} - \widetilde{\mathbf{W}}\mathbf{X})\mathbf{X}^\top\right] \odot \frac{\partial\sigma(\mathbf{R})}{\partial\mathbf{R}}. \tag{29}$$

For the regularization term $f_{reg}$, its gradient is expressed directly in differential form without expanding the chain rule:

$$\frac{\partial\mathcal{L}_{reg}}{\partial\mathbf{R}} = \lambda \cdot \frac{\partial f_{reg}(\mathbf{R})}{\partial\mathbf{R}}. \tag{30}$$

The total gradient of the loss with respect to $\mathbf{R}$ combines both components:

$$\frac{\partial \mathcal{L}}{\partial \mathbf{R}} = \frac{\partial \mathcal{L}_{recon}}{\partial \mathbf{R}} + \frac{\partial \mathcal{L}_{reg}}{\partial \mathbf{R}}. \tag{31}$$

That is the gradient for LLR, and we will next compare it to the gradient used in the learnable rounding process for symmetric linear quantization counterpart. While the loss function remains the same, the soft-quantized weights in the linear quantization setting are defined as:

$$Quant : \mathbf{Q_W}' = \text{clamp}\left(\left\lceil (2^{N-1} - 1) \cdot \frac{|\mathbf{W}|}{s'} \right\rceil - \sigma\left(\mathbf{R}'\right), 0, 2^{N-1} - 1\right), \quad s' = \max(|\mathbf{W}|), \tag{32}$$

$$Dequant : \widetilde{\mathbf{W}}' = \frac{s'}{2^{N-1} - 1} \cdot \text{sgn}(\mathbf{W}) \odot \mathbf{Q_W}'. \tag{33}$$

The gradient of $\mathcal{L}'$ with respect to $\mathbf{R}'$ is derived as:

$$\frac{\partial \mathcal{L}'_{recon}}{\partial \mathbf{R}'} = \frac{2s'}{2^{N-1} - 1} \cdot \mathbf{M}'_c \odot \text{sgn}(\mathbf{W}) \odot \left[(\mathbf{W}\mathbf{X} - \widetilde{\mathbf{W}}'\mathbf{X})\mathbf{X}^{\top}\right] \odot \frac{\partial \sigma(\mathbf{R}')}{\partial \mathbf{R}'}, \tag{34}$$

$$\frac{\partial \mathcal{L}'_{reg}}{\partial \mathbf{R}'} = \lambda \cdot \frac{\partial f_{reg}(\mathbf{R}')}{\partial \mathbf{R}'}, \tag{35}$$

$$\frac{\partial \mathcal{L}'}{\partial \mathbf{R}'} = \frac{\partial \mathcal{L}'_{recon}}{\partial \mathbf{R}'} + \frac{\partial \mathcal{L}'_{reg}}{\partial \mathbf{R}'}. \tag{36}$$

The key difference between the gradients of logarithmic and linear learnable rounding lies in the term $\partial \mathcal{L}_{recon}/\partial \mathbf{R}$, which originates from their distinct quantization strategies. In logarithmic quantization, the reconstruction gradient derived in Eq. (29) includes an exponential term $2^{-\mathbf{Q_W}}$ and a logarithmic scaling factor $s \ln 2$. In contrast, linear quantization uses a constant scaling factor of $\frac{s'}{2^{N-1}-1}$ in Eq. (34). The presence of the exponential term $2^{-\mathbf{Q_W}}$ in LLR results in smaller gradients for weights with small magnitudes, while the constant scaling in linear quantization provides more uniform gradient flow across all weight magnitudes. Assuming $s' \approx s$, the ratio between the gradient scaling factors in logarithmic and linear quantization becomes $\ln 2 \cdot 2^{-\mathbf{Q_W}}$ to $\frac{1}{2^{N-1}-1}$. This ratio is particularly useful when considering the appropriate choice of the regularization coefficient $\lambda$, which governs the trade-off between reconstruction fidelity and rounding determinism. A larger $\lambda$ encourages harder rounding behavior, while a smaller $\lambda$ allows softer updates during optimization.

# B  SYNERGISTIC EFFECTS OF MULTI-LEVEL OHS AND LLR

We conduct a detailed analysis of the synergistic relationship between the proposed OHS and LLR techniques, drawing insights from the comprehensive ablation results in Table 11. Our findings reveal that their integration is a key driver of LogART effectiveness: beyond being merely additive in quantization performance, the synergy between OHS and LLR also accelerates convergence and substantially enhances overall optimization.

## B.1  OHS AS AN ENABLER FOR EFFECTIVE LEARNABLE ROUNDING

The effectiveness of LLR is ultimately bounded by the quality of the underlying quantization grid. While LLR can optimize how values are mapped to existing quantization points, it cannot compensate for a poorly chosen grid. The final accuracy will remain limited regardless of rounding quality. This is where synergy with OHS becomes critical. The multi-level OHS (comprising tensor-wise ABS, block-wise SFS, and block-wise DBS) first establishes an optimal quantization grid, which then enables LLR to fully exploit its rounding optimization.

ABS corrects for asymmetry based on the original weight distribution and therefore requires only tensor-wise search. SFS improves robustness to outliers, while DBS selects the most appropriate base. Both SFS and DBS operate in an activation-aware manner through block-wise search. This multi-level search strategy also avoids the vast search space that would otherwise prolong OHS runtime. By constructing a better-structured grid, OHS enables LLR to converge to a superior final solution with lower overall quantization error. This is validated by the consistent and cumulative performance gains in Table 11, where each OHS component added on top of the LLR baseline further improves results, resulting in the best performance when all components are active.

## B.2  LLR AS A FINER-GRAINED OPTIMIZATION BUILT ON EFFECTIVE SEARCH

Compared to the coarse-grained OHS, which focuses on optimizing quantization hyperparameters for an entire weight tensor, LLR operates at the element level. Specifically, LLR learns the optimal rounding decision for each weight element to minimize the task-aware loss. The additive benefit of LLR is clearly demonstrated in Table 11, where, under every configuration of DBS, SFS, and ABS, the inclusion of LLR consistently leads to higher accuracy than the corresponding setting without it.

This is because a naive RTN policy produces a highly suboptimal and rugged loss landscape. LLR fundamentally changes this dynamic by introducing a task-aware, learnable rounding mechanism. A drawback of LLR is that it requires backpropagation at the layer or block level, making the gradient descent process resource-intensive and time-consuming, particularly for large models such as LLMs. Therefore, improving the convergence speed of LLR has become a critical topic.

## B.3  THEORETICAL ANALYSIS OF COMPONENT SYNERGY

We mathematically justify the synergy between OHS and LLR by decomposing the quantization error in the task-aware metric space. The LogART optimization objective minimizes the task-aware reconstruction loss. Let $\widetilde{\mathbf{W}}$ denote the soft-quantized weight matrix and $\Delta\mathbf{W} = \mathbf{W} - \widetilde{\mathbf{W}}$ be the quantization error. The loss function can be expressed as a Frobenius norm by incorporating the matrix square root of the Hessian $\mathbf{H} = \mathbb{E}[\mathbf{X}\mathbf{X}^\top]$ directly into the norm:

$$\mathcal{L} = \mathrm{tr}\left(\Delta\mathbf{W}\mathbf{H}\Delta\mathbf{W}^\top\right) = \left\|\Delta\mathbf{W}\mathbf{H}^{\frac{1}{2}}\right\|_F^2. \tag{37}$$

The triangle inequality establishes an upper limit on the quantization error norm, separating it into two distinct geometric terms:

$$\left\|\Delta\mathbf{W}\mathbf{H}^{\frac{1}{2}}\right\|_F^2 \le \Big(\underbrace{\left\|(\mathbf{W} - \Pi_{\mathcal{C}(\theta)}^{\mathbf{H}}(\mathbf{W}))\mathbf{H}^{\frac{1}{2}}\right\|_F}_{\mathcal{E}_1 \text{ (OHS)}} + \underbrace{\left\|(\Pi_{\mathcal{C}(\theta)}^{\mathbf{H}}(\mathbf{W}) - \widetilde{\mathbf{W}})\mathbf{H}^{\frac{1}{2}}\right\|_F}_{\mathcal{E}_2 \text{ (LLR)}}\Big)^2. \tag{38}$$

Here, $\mathcal{C}(\theta)$ denotes the quantization codebook characterized by OHS hyperparameters $\theta = \{s_{\mathrm{of}}, n_1, l_a\}$, and $\Pi_{\mathcal{C}(\theta)}^{\mathbf{H}}(\mathbf{W})$ represents the ideal projection of $\mathbf{W}$ onto $\mathcal{C}(\theta)$ under the Hessian-weighted metric. Since minimizing a non-negative norm $\mathcal{E}_1/\mathcal{E}_2$ is equivalent to minimizing its square

$\mathcal{E}_1^2/\mathcal{E}_2^2$, we formulate the following optimization objectives in the squared domain for mathematical convenience and consistency with the loss definition. The term $\mathcal{E}_1$ defines the intrinsic discretization error determined purely by the geometry of the codebook. OHS explicitly minimizes this term to identify the optimal $\theta^*$:

$$\theta^* = \arg\min_{\theta} \sum_{i,j} \min_{c \in \mathcal{C}(\theta)} \left( (\mathbf{W}_{ij} - c)^2 \cdot \mathbf{H}_{jj} \right), \tag{39}$$

where $\theta^*$ denotes the optimal set of hyperparameters found by OHS, $c$ denotes a codebook value, and $\mathbf{H}_{jj}$ refers to the diagonal elements of the Hessian. Specifically, SFS aligns the global scale through $s_{\text{of}}$, DBS optimizes local density using $n_1$, and ABS adjusts support bounds by $l_a$. The term $\mathcal{E}_2$ represents the optimization gap between the ideal projection $\Pi^{\mathbf{H}}_{\mathcal{C}(\theta)}$ and the result $\widetilde{\mathbf{W}}$ determined by LLR. LLR learns the optimal rounding variable $\mathbf{R}^*$:

$$\mathbf{R}^* = \arg\min_{\mathbf{R}} \left\| (\Pi^{\mathbf{H}}_{\mathcal{C}(\theta^*)}(\mathbf{W}) - \widetilde{\mathbf{W}})\mathbf{H}^{\frac{1}{2}} \right\|_F^2 \quad \text{s.t.} \quad \widetilde{\mathbf{W}} \in \mathcal{C}(\theta^*). \tag{40}$$

This mathematical decomposition proves the synergistic relationship. OHS searches for the optimal quantization grid that minimizes the intrinsic discretization error, while LLR learns the optimal rounding decision to minimize the residual error on that established grid. Consequently, the full integration of all proposed components yields the theoretically optimal configuration.

### B.4 ACCELERATED CONVERGENCE

The synergy between OHS and LLR also accelerates convergence during the learnable rounding phase. When LLR operates on a poorly configured grid, its optimization must compensate for the inherent flaws of that grid. By using OHS to establish a more suitable quantization grid, the rounding problem becomes simpler and better-posed, allowing the optimizer to reach a high-quality solution in fewer iterations.

Table 7: Performance (PPL) and runtime comparison of LLR convergence with and without OHS.

| ABS | SFS | DBS | LLR iters | Runtime (min) | | | PPL | |
| --- | --- | --- | --- | --- | --- | --- | --- | --- |
| | | | | OHS | LLR | Total | WikiText-2 | C4 |
| × | × | × | 2000 | 0.05 | 3.95 | 4.00 | 36.27 | 34.60 |
| ✓ | ✓ | ✓ | 500 | 0.28 | 0.97 | 1.25 | 31.15 | 30.44 |

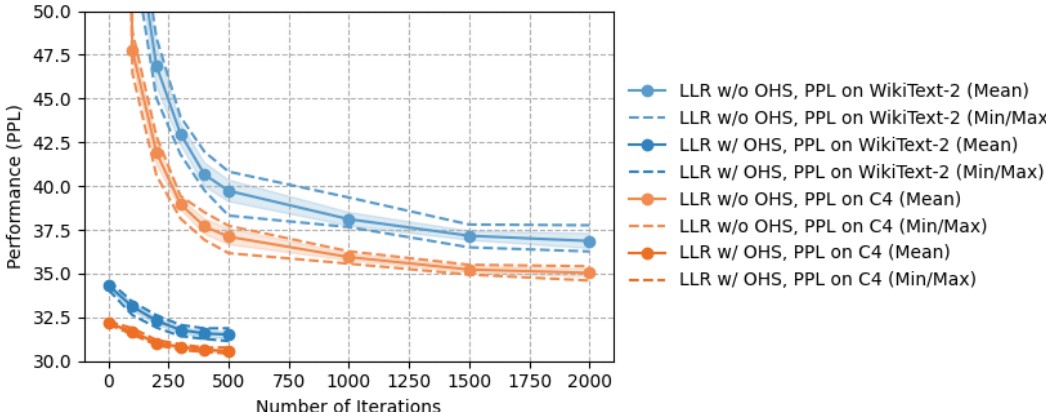

Figure 5: LLR combined with multi-level OHS converges faster and achieves higher accuracy.

To validate the accelerated convergence, we conducted a comparative experiment on the OPT-125M model using the WikiText-2 calibration dataset. We compare two configurations, both including LLR. The experimental group employs our full multi-level OHS with the same settings as our main LLM experiments. The control group uses a simplified strategy where ABS, SFS, and DBS are all disabled,

with the total number of iterations, learning rate and weight of the rounding loss to 2000, 0.015, and 1, respectively. The results, shown in Table 7 and Figure 5, indicate that while multi-level OHS itself incurs longer runtime due to the required local forward computations, it enables LLR to converge more efficiently to a better local minimum, resulting in superior overall accuracy and total runtime.

## C  HAF MODULE AND HARDWARE-LEVEL ANALYSIS

We analyze the hardware complexity of the AE designs for the core $\widehat{\mathbf{W}}\mathbf{X}$ computation resulting from different PTQ methods applied to weights:

- **Linear**: Linear quantization schemes include symmetric and asymmetric approaches. Symmetric linear PTQ allows the AE to be implemented using low-bitwidth fixed-point signed integer multipliers (INT SMUL). Asymmetric linear PTQ, due to the introduction of zero-point offsets in quantized weights, typically requires INT SMUL in Figure 4(a) with higher bitwidth or additional hardware to handle these offsets.
- **Log2**: Base-2 logarithmic PTQ offers significant hardware advantages by enabling the replacement of multipliers with shifters, as shown in Figure 4(b). To accommodate the sign of the weights, the AE incorporates signed operational logic to compute correctly.
- **Log$\sqrt{2}$**: Base-$\sqrt{2}$ quantization does not directly translate to multiplier-free operations. However, hardware efficiency can be achieved by $\sqrt{2} \approx 2^0 + 2^{-1}$ Xu et al. (2018; 2023). This approximation allows multipliers to be replaced with shift-add operations, as shown in Figure 4(c).
- **AdaLog**: It is important to note that AdaLog Wu et al. (2024) does not support logarithmic quantization for weights. Instead, it applies logarithmic quantization to activations while keeping weights linear. To enable flexible logarithmic bases, AdaLog adopts an AE design that combines LUTs, a multiplier, and a shifter, as illustrated in Figure 4(d). This design provides flexibility but at the cost of higher resource utilization.
- **LogART**: LogART adopts a multiplier-free AE design (Figure 4(e)) that facilitates DLog-quantized $\widehat{\mathbf{W}}\mathbf{X}$ computation through a shift-add approach enabled by HAF. Configured with dynamic logarithmic bases, the AE takes in the quantized weight and input activation codes, along with control signals such as $n_1$ and $chk\_even$. The decoder, implementable with simple combinational logic, generates the enable signal for the Approx module and the shifting bits for the Shift module.

To effectively incorporate the hardware approximation into the LogART LLR process, allowing the induced error to be absorbed as noise during the learning process, we introduce an add-on HAF into the quantized forward pass. As described in Eq. (9)-(17), the floating-point weight matrix $\mathbf{W}$ is first quantized into its integer-form representation $\mathbf{Q_W}$, and subsequently dequantized back into hard-quantized $\widehat{\mathbf{W}}$. To model the hardware approximation, an element-wise mask $\mathbf{M}$ is generated, which is used to modify the dequantized weights $\widehat{\mathbf{W}}$, resulting in an approximated version $\widehat{\mathbf{W}}'$:

$$\mathbf{M} = (\mathbf{Q} \bmod 2) \odot \left[\mathbf{B} = \sqrt{2}\right], \quad \widehat{\mathbf{W}}' = \widehat{\mathbf{W}} \odot (\mathbf{1} + (\gamma - 1)\mathbf{M}), \tag{41}$$

where $\bmod$ denotes the modulo operation that returns the remainder after division, $\left[\mathbf{B} = \sqrt{2}\right]$ is an element-wise indicator function identifying which elements of $\mathbf{B}$ are equal to $\sqrt{2}$, and $\gamma$ represents the hardware approximation factor:

$$\gamma = \frac{\mathrm{SDE}(\sqrt{2}, K)}{\sqrt{2}}. \tag{42}$$

Here, $\mathrm{SDE}(\sqrt{2}, K)$ denotes the Signed Dyadic Expansion function, which takes the real number $\sqrt{2}$ and an integer $K$, and returns a signed dyadic expansion approximation with $K$ terms:

$$\mathrm{SDE}(\sqrt{2}, K) = \sum_{k=1}^{K} a_k \cdot \frac{1}{2^{d_k}}, \quad \text{where } a_k \in \{-1, +1\}, \ d_k \in \mathbb{N}, \ d_1 < d_2 < d_3 < \cdots. \tag{43}$$

This expansion is useful for replacing the multipliers required in $\sqrt{2}$-involved computation with shifters and adders, thereby enabling efficient hardware AE design. The construction of the SDE function follows a greedy iterative approach. Starting with the initial residual $r_1 = \sqrt{2}$, at each iteration $k$, we search for the integer $d_k$ that meet the requirement:

$$\frac{1}{2^{d_k}} \leq |r_k| < \frac{1}{2^{d_k-1}}. \tag{44}$$

We then assign the sign $a_k$ to match the sign of the residual:

$$a_k = \text{sgn}(r_k) \in \{-1, +1\}. \tag{45}$$

Next, the selected dyadic term is subtracted from the residual:

$$r_{k+1} = r_k - a_k \cdot \frac{1}{2^{d_k}} \tag{46}$$

This process is repeated to generate a sequence of signed dyadic terms that bring the residual closer to zero. After $K$ steps, the partial sum $\sum_{k=1}^{K} a_k \cdot \frac{1}{2^{d_k}}$ provides an approximation of $\sqrt{2}$ with $K$ terms. By determining the number of terms $K$, we can generate the corresponding AE design and obtain the SDE approximation of $\sqrt{2}$. The absolute error introduced to the approximation of $\sqrt{2}$ by HAF is defined as:

$$\epsilon_K = \left| \sqrt{2} - \text{SDE}(\sqrt{2}, K) \right| = |r_{K+1}| = \left| r_K - a_K \cdot \frac{1}{2^{d_K}} \right|. \tag{47}$$

From Eq. (44) and (45), we derive the theoretical upper bound of $\epsilon_K$ as:

$$\epsilon_K = |r_K| - \frac{1}{2^{d_K}}, \tag{48}$$

$$< \frac{1}{2^{d_K - 1}} - \frac{1}{2^{d_K}} = \frac{1}{2^{d_K}}. \tag{49}$$

Utilizing this approximation to compute $\gamma$ and $\widehat{\mathbf{W}}'$, the error induced by HAF on the quantized weights $\widehat{\mathbf{W}}$ is given by:

$$\Delta\widehat{\mathbf{W}} = \widehat{\mathbf{W}}' - \widehat{\mathbf{W}} = \widehat{\mathbf{W}} \odot (\gamma - 1)\mathbf{M}, \tag{50}$$

where the hardware approximation factor $\gamma$ relates to the final residual $r_{K+1}$ as:

$$\gamma = \frac{\text{SDE}(\sqrt{2}, K)}{\sqrt{2}} = \frac{\sqrt{2} - r_{K+1}}{\sqrt{2}} = 1 - \frac{r_{K+1}}{\sqrt{2}}. \tag{51}$$

Applying the $\sqrt{2}$ error bound in Eq. (49) yields the theoretical upper bound of the weight error:

$$\left| \Delta\widehat{\mathbf{W}} \right| = \left| \widehat{\mathbf{W}} \right| \odot \frac{|r_{K+1}|}{\sqrt{2}} \odot \mathbf{M} \tag{52}$$

$$< \left| \widehat{\mathbf{W}} \right| \odot \frac{1}{\sqrt{2} \cdot 2^{d_K}} \odot \mathbf{M}. \tag{53}$$

This local weight error propagates through the quantized forward pass, affecting the task-aware reconstruction loss $\mathcal{L}$:

$$\mathbb{E}\left[ \mathcal{L}\left( \Delta\widehat{\mathbf{W}} \right) \right] = \mathbb{E}\left[ \left\| \Delta\widehat{\mathbf{W}}\mathbf{X} \right\|_F^2 \right] = \text{tr}(\Delta\widehat{\mathbf{W}}\mathbf{H}\Delta\widehat{\mathbf{W}}^\top), \quad \mathbf{H} = \mathbb{E}\left[ \mathbf{X}\mathbf{X}^\top \right], \tag{54}$$

$$= \frac{r_{K+1}^2}{2} \cdot \text{tr}\left( (\widehat{\mathbf{W}} \odot \mathbf{M}) \cdot \mathbf{H} \cdot (\widehat{\mathbf{W}} \odot \mathbf{M})^\top \right), \tag{55}$$

$$< \frac{1}{2^{1+2d_K}} \cdot \text{tr}\left( (\widehat{\mathbf{W}} \odot \mathbf{M}) \cdot \mathbf{H} \cdot (\widehat{\mathbf{W}} \odot \mathbf{M})^\top \right). \tag{56}$$

This derivation provides the theoretical upper bound for the HAF-induced error in the reconstruction loss, which diminishes exponentially with respect to $d_K$. While a larger $K$ reduces reconstruction loss and potentially improves accuracy, it incurs increasing hardware overhead. Specifically, as $K$ increases, the area and power consumption of the AE design grow sublinearly.

To resolve this trade-off, we integrate the HAF-induced error directly into the LLR learning process, allowing the model to compensate via element-wise optimized rounding decisions. During backpropagation, HAF introduces an additional gradient path from $\widehat{\mathbf{W}}'$ to $\widehat{\mathbf{W}}$:

$$\frac{\partial\widehat{\mathbf{W}}'}{\partial\widehat{\mathbf{W}}} = \mathbf{1} + (\gamma - 1)\mathbf{M}. \tag{57}$$

By incorporating this error into the gradient descent of LLR, the optimization theoretically learns to absorb this deviation. This ensures a dual benefit. The final accuracy using HAF ($K = 2$) is

Table 8: Evaluation of how $K$ affects top-1 accuracy (%) and GPU runtime for 4-bit PTQ on CNNs.

| Method | HAF | | ResNet18 | ResNet50 | MobileNetV2 |
|---|---|---|---|---|---|
| LogART | - | Acc | 70.79 | 76.57 | 71.62 |
| | $\times$ | Runtime (min) | 1.2 | 3.1 | 2.6 |
| $K = 2$ | $\times$ | Acc | 68.75 | 74.41 | 62.64 |
| $K = 3$ | $\times$ | Acc | 70.68 | 76.36 | 70.94 |
| $K = 4$ | $\times$ | Acc | 70.59 | 76.48 | 71.49 |
| | $\checkmark$ | Runtime (min) | 1.3 | 3.4 | 2.8 |
| $K = 2$ | $\checkmark$ | Acc | 70.71 | 76.48 | 71.45 |
| $K = 3$ | $\checkmark$ | Acc | 70.74 | 76.59 | 71.63 |
| $K = 4$ | $\checkmark$ | Acc | 70.79 | 76.53 | 71.64 |

Table 9: Evaluation of how $K$ affects top-1 accuracy (%) and GPU runtime for 3/4-bit PTQ on vision tranformers.

| Method | W | HAF | | ViT-S | ViT-B | DeiT-T | DeiT-B |
|---|---|---|---|---|---|---|---|
| LogART | 4-bit | - | Acc | 81.06 | 85.02 | 71.62 | 81.92 |
| | 4-bit | $\times$ | Runtime (min) | 6.7 | 10.9 | 5.6 | 10.9 |
| $K = 2$ | 4-bit | $\times$ | Acc | 80.12 | 84.36 | 70.14 | 81.51 |
| $K = 3$ | 4-bit | $\times$ | Acc | 80.81 | 84.89 | 71.35 | 81.79 |
| $K = 4$ | 4-bit | $\times$ | Acc | 80.97 | 84.96 | 71.50 | 81.84 |
| | 4-bit | $\checkmark$ | Runtime (min) | 6.8 | 11.4 | 5.7 | 11.2 |
| $K = 2$ | 4-bit | $\checkmark$ | Acc | 81.02 | 84.99 | 71.59 | 81.88 |
| $K = 3$ | 4-bit | $\checkmark$ | Acc | 80.98 | 85.01 | 71.70 | 81.94 |
| $K = 4$ | 4-bit | $\checkmark$ | Acc | 80.98 | 84.98 | 71.63 | 81.92 |
| LogART | 3-bit | - | Acc | 79.56 | 84.54 | 70.21 | 81.51 |
| | 3-bit | $\times$ | Runtime (min) | 5.9 | 9.9 | 5.0 | 10.1 |
| $K = 2$ | 3-bit | $\times$ | Acc | 78.89 | 84.18 | 69.50 | 81.28 |
| $K = 3$ | 3-bit | $\times$ | Acc | 79.35 | 84.36 | 69.80 | 81.39 |
| $K = 4$ | 3-bit | $\times$ | Acc | 79.41 | 84.42 | 70.10 | 81.44 |
| | 3-bit | $\checkmark$ | Runtime (min) | 6.0 | 10.3 | 5.2 | 10.3 |
| $K = 2$ | 3-bit | $\checkmark$ | Acc | 79.67 | 84.58 | 70.14 | 81.45 |
| $K = 3$ | 3-bit | $\checkmark$ | Acc | 79.50 | 84.56 | 70.25 | 81.50 |
| $K = 4$ | 3-bit | $\checkmark$ | Acc | 79.46 | 84.51 | 70.26 | 81.52 |

negligibly different from an ideal $\sqrt{2}$ implementation, while the LogART AE maintains its most efficient configuration. To assess the trade-off and the effect of $K$ on final accuracy, we conduct 3-bit and 4-bit per-channel weight quantization experiments on LLM, CNN and vision transformer models.

The experimental setup for LogART follows the configuration in Section 4.2, enabling full OHS in combination with LLR. An "ideal" LogART variant, which assumes a perfect representation of $\sqrt{2}$, is used as the benchmark. To evaluate the impact of hardware approximation, we compare the top-1 accuracy when applying the approximation function with:

$$K = 2 : \quad \sqrt{2} \approx 1 + \frac{1}{2} \tag{58}$$

$$K = 3 : \quad \sqrt{2} \approx 1 + \frac{1}{2} - \frac{1}{2^4} \tag{59}$$

$$K = 4 : \quad \sqrt{2} \approx 1 + \frac{1}{2} - \frac{1}{2^4} - \frac{1}{2^6} \tag{60}$$

Experimental results in Tables 10, 8, and 9 confirm the effectiveness of the proposed HAF. A naive hardware approximation presents a difficult trade-off: increasing the number of approximation terms ($K$) improves accuracy by providing a better estimate of $\sqrt{2}$, but also significantly increases the area and power consumption of the LogART AE. The proposed HAF resolves this dilemma. Experimental results show that the accuracies with HAF are consistently higher than those under

Table 10: Evaluation of how $K$ affects 3-bit PTQ on OPT-125M. (Calibration data from WikiText-2)

| Method | HAF | | | WikiText-2 | C4 |
|---|---|---|---|---|---|
| LogART | - | PPL | | 31.15 | 30.44 |
| | $\times$ | Runtime (s) | | 75.1 | |
| $K = 2$ | $\times$ | PPL | | 31.94 | 30.66 |
| $K = 3$ | $\times$ | PPL | | 31.62 | 30.58 |
| $K = 4$ | $\times$ | PPL | | 31.48 | 30.54 |
| | $\checkmark$ | Runtime (s) | | 79.5 | |
| $K = 2$ | $\checkmark$ | PPL | | 31.30 | 30.40 |
| $K = 3$ | $\checkmark$ | PPL | | 31.34 | 30.42 |
| $K = 4$ | $\checkmark$ | PPL | | 31.30 | 30.44 |

naive hardware approximation. By incorporating the hardware approximation error into the OHS and LLR optimization process, the chosen hyperparameters and element-wise rounding decisions become robust to the hardware approximation error and the choice of $K$. The performance degradation is negligible, and in some cases, the HAF-enabled model can even surpass the "ideal" LogART (e.g., with $K = 2$ for 3-bit quantization on the ViT-Small model). As HAF mitigates the accuracy impact of a lower $K$, we select $K = 2$ for our final LogART AE design, achieving superior hardware efficiency while maintaining performance.

# D  FULL ABLATION STUDY

We provide comprehensive component-by-component results of the full ablation study, which are excluded from the main text due to page limitations. The evaluation encompasses LLMs (OPT-125M, LLaMA2-7B), CNNs (ResNet18, MobileNetV2), and vision transformers (DeiT-Tiny, ViT-Base). These experiments validate the individual and synergistic contributions of the four key components of LogART: DBS, SFS, ABS, and LLR.

To clarify the ablation study conditions in Table 11 and Table 12, the following settings are used when components are disabled: without LLR, a standard RTN policy is applied; without ABS, a symmetric quantizer is used; without SFS, the scaling factor is fixed to 1; and without DBS, a fixed Log2 base is used. The absolute baseline (first row), with all components disabled, represents a symmetric, single-base Log2 quantizer that uses a simple RTN policy and a fixed scaling factor of 1.

Table 11: Full ablation results of LogART key components on LLMs with 3-bit channel-wise weight quantization, evaluated in terms of calibration data (from WikiText-2) dependency, PPL on WikiText-2 dataset, time cost, and GPU memory cost.

| DBS | SFS | ABS | LLR | Calib. Data | OPT-125M | | | LLaMA2-7B | | |
|---|---|---|---|---|---|---|---|---|---|---|
| | | | | | PPL | Time | Memory | PPL | Time | Memory |
| × | × | × | × | - | 170.64 | 0.7 s | 0.40 GB | 60.16 | 13.0 s | 9.8 GB |
| × | × | × | ✓ | 32 | 38.55 | 61.3 s | 0.75 GB | 9.74 | 58.6 min | 20.9 GB |
| × | × | ✓ | × | - | 79.70 | 0.7 s | 0.40 GB | 8.28 | 13.2 s | 9.8 GB |
| × | × | ✓ | ✓ | 32 | 36.39 | 61.3 s | 0.75 GB | 6.44 | 58.2 min | 20.9 GB |
| × | ✓ | × | × | 32 | 38.41 | 12.1 s | 0.75 GB | 6.66 | 6.6 min | 20.9 GB |
| × | ✓ | × | ✓ | 32 | 33.21 | 64.6 s | 0.75 GB | 6.24 | 63.1 min | 20.9 GB |
| × | ✓ | ✓ | × | 32 | 35.15 | 12.2 s | 0.75 GB | 6.55 | 6.6 min | 20.9 GB |
| × | ✓ | ✓ | ✓ | 32 | 32.55 | 64.6 s | 0.75 GB | 6.23 | 63.5 min | 20.9 GB |
| ✓ | × | × | × | 32 | 66.63 | 3.8 s | 0.75 GB | 18.49 | 83.2 s | 20.9 GB |
| ✓ | × | × | ✓ | 32 | 35.46 | 62.7 s | 0.75 GB | 9.26 | 59.0 min | 20.9 GB |
| ✓ | × | ✓ | × | 32 | 47.92 | 3.8 s | 0.75 GB | 7.82 | 82.2 s | 20.9 GB |
| ✓ | × | ✓ | ✓ | 32 | 33.68 | 62.9 s | 0.75 GB | 6.38 | 59.1 min | 20.9 GB |
| ✓ | ✓ | × | × | 32 | 36.10 | 16.8 s | 0.75 GB | 6.56 | 17.9 min | 20.9 GB |
| ✓ | ✓ | × | ✓ | 32 | 32.37 | 75.0 s | 0.75 GB | 6.19 | 73.7 min | 20.9 GB |
| ✓ | ✓ | ✓ | × | 32 | 34.29 | 17.0 s | 0.75 GB | 6.45 | 17.9 min | 20.9 GB |
| ✓ | ✓ | ✓ | ✓ | 32 | 31.15 | 75.1 s | 0.75 GB | 6.14 | 74.2 min | 20.9 GB |

Table 12: Full ablation results of LogART key components on CNN and vision transformer models with 4-bit channel-wise weight quantization, evaluated in terms of top-1 accuracy on the ImageNet dataset, and GPU runtime.

| DBS | SFS | ABS | LLR | ResNet18 | | MobileNetV2 | | ViT-B | | DeiT-T | |
|---|---|---|---|---|---|---|---|---|---|---|---|
| | | | | Acc(%) | Time | Acc(%) | Time | Acc(%) | Time | Acc(%) | Time |
| × | × | × | × | 31.53 | 0.9 s | 1.22 | 1.9 s | 79.55 | 0.0 min | 57.25 | 0.0 min |
| × | × | × | ✓ | 69.75 | 60.8 s | 68.58 | 133.0 s | 84.43 | 8.1 min | 69.58 | 4.2 min |
| × | × | ✓ | × | 32.04 | 1.0 s | 1.25 | 1.9 s | 79.63 | 0.0 min | 57.37 | 0.0 min |
| × | × | ✓ | ✓ | 69.78 | 61.0 s | 68.62 | 133.4 s | 84.50 | 8.1 min | 69.65 | 4.2 min |
| × | ✓ | × | × | 67.13 | 2.8 s | 58.43 | 5.6 s | 83.25 | 0.6 min | 64.32 | 0.3 min |
| × | ✓ | × | ✓ | 69.82 | 62.7 s | 68.96 | 138.8 s | 84.59 | 8.9 min | 69.81 | 4.5 min |
| × | ✓ | ✓ | × | 67.29 | 2.8 s | 58.48 | 5.6 s | 83.31 | 0.6 min | 64.38 | 0.3 min |
| × | ✓ | ✓ | ✓ | 69.89 | 62.8 s | 69.02 | 138.9 s | 84.67 | 8.9 min | 69.85 | 4.5 min |
| ✓ | × | × | × | 68.45 | 1.0 s | 66.91 | 2.3 s | 84.24 | 0.1 min | 69.17 | 0.0 min |
| ✓ | × | × | ✓ | 70.50 | 61.7 s | 71.24 | 134.4 s | 84.91 | 8.3 min | 71.44 | 4.2 min |
| ✓ | × | ✓ | × | 69.04 | 1.0 s | 67.00 | 2.3 s | 84.28 | 0.1 min | 69.32 | 0.0 min |
| ✓ | × | ✓ | ✓ | 70.59 | 61.5 s | 71.25 | 134.6 s | 84.94 | 8.3 min | 71.48 | 4.3 min |
| ✓ | ✓ | × | × | 69.69 | 10.8 s | 69.47 | 22.0 s | 84.61 | 2.8 min | 70.29 | 1.4 min |
| ✓ | ✓ | × | ✓ | 70.69 | 71.1 s | 71.55 | 153.4 s | 84.99 | 10.9 min | 71.56 | 5.6 min |
| ✓ | ✓ | ✓ | × | 69.89 | 11.6 s | 69.86 | 24.4 s | 84.67 | 2.8 min | 70.40 | 1.4 min |
| ✓ | ✓ | ✓ | ✓ | 70.79 | 72.3 s | 71.62 | 156.6 s | 85.02 | 10.9 min | 71.62 | 5.6 min |

The full ablation results clearly illustrate the synergistic and cumulative effects of the LogART components. While each module provides a benefit in isolation, they are most effective when combined. A clear observation from Table 11 and Table 12 is that regardless of the starting component or any combination of components, adding an additional module always leads to further improvement in accuracy (evidenced by lower PPL on LLMs or higher top-1 accuracy on ImageNet). This phenomenon highlights the complementary nature of our proposed modules, with each one targeted to address a distinct challenge in logarithmic PTQ.

We further analyze the trade-off between the performance contribution and computational efficiency of each component. DBS serves as the critical foundation for logarithmic PTQ, offering the best efficiency-to-performance ratio. SFS demonstrates strong standalone robustness, particularly for LLMs, restoring LLaMA2-7B from 60.16 PPL to 6.66. ABS functions best as a complementary refinement, yielding consistent accuracy improvements with negligible GPU runtime and memory overhead. Finally, LLR, operating at the finest element-wise granularity, stands out as a high-precision powerhouse but comes at the cost of computational time.

The effectiveness of each component varies across model architectures. Compact models like MobileNetV2 are vulnerable to quantization noise (1.22% baseline), making DBS critical for restoring usable accuracy (66.91%). In contrast, large-scale architectures like vision transformers and LLMs exhibit greater resilience to initial quantization errors. Notably, SFS has a more significant impact on LLMs compared to CNNs and vision transformers, likely due to the widely observed outliers in language models. ABS yields the most substantial gains on LLaMA2-7B, effectively addressing its asymmetric weight distribution. For all models to restore near full-precision performance, LLR is necessary for fine-grained refinement.

In summary, the ablation study validates the design of LogART as a unified framework. The lightweight components (DBS, SFS, ABS) ensure robust functionality and efficient basis construction, while LLR provides the necessary precision refinement. Combined, they yield the best accuracy performance.

# E    IMPACT OF THE CALIBRATION DATASET ON PERFORMANCE

## E.1    IMPACT OF CALIBRATION DATASET SOURCE

The main experiments on LLMs, detailed in Table 3, utilize the C4 dataset for calibration to ensure a fair comparison with prior SOTA methods. This section evaluates the robustness of LogART by measuring its sensitivity to the calibration data source. We conduct a parallel set of experiments where models are calibrated using the WikiText-2 dataset instead of C4. All other experimental conditions, including the models and experimental settings, remain identical to those in the main text to isolate the effect of the calibration data. The results of this comparison are summarized in Table 13.

Table 13: Comparison of performance (PPL) for per-channel 3-bit LogART on LLM weights using 32 random 2048-token segments for calibration from the C4 and WikiText-2 datasets.

| Calib. Dataset | | OPT-125M | OPT-1.3B | OPT-6.7B | LLaMA2-7B | LLaMA3-8B |
|---|---|---|---|---|---|---|
| C4 | PPL (WikiText-2) | 31.52 | 15.53 | 11.11 | 6.31 | 8.19 |
| | PPL (C4) | 29.98 | 17.29 | 13.37 | 8.38 | 12.44 |
| WikiText-2 | PPL (WikiText-2) | 31.15 | 15.61 | 11.37 | 6.14 | 7.83 |
| | PPL (C4) | 30.44 | 17.60 | 13.54 | 8.55 | 13.27 |

Table 13 reveals a distinct in-domain alignment pattern. Models generally achieve lower PPL when the calibration source matches the test domain. This indicates that the calibration set helps the quantization parameters adapt to the specific linguistic distribution of the target domain. The performance difference stems from domain alignment and outlier coverage. C4 provides high linguistic diversity, covering a wider range of activation outliers, which ensures better generalization. In contrast, WikiText-2 offers a highly consistent, formal distribution, leading to slightly lower PPL due to in-domain overfitting when used for both calibration and test.

Crucially, despite these variations, LogART exhibits strong robustness. The performance variance across different calibration sources is marginal, and the model maintains high accuracy regardless of the source. For general-purpose deployment and fair SOTA comparison, we recommend using a large-scale, diverse dataset like C4. However, for tasks targeting a specific domain, utilizing in-domain calibration data yields the best fine-grained performance.

## E.2    IMPACT OF CALIBRATION DATASET SIZE

In addition to the choice of calibration data source, the size of the calibration dataset is also a critical factor for the practicality and efficiency of a PTQ method. A method that requires a large number of samples can be costly and time-consuming. In this section, we evaluate the sensitivity of LogART performance to the number of calibration samples. The study is conducted on the OPT-125M and LLaMA2-7B models with 3-bit per-channel weight quantization, using the calibration data from the WikiText-2 dataset. We vary the number of calibration segments, testing sizes of 32 and 128 samples (each containing 2048 tokens), and measure the resulting PPL. The results for OHS alone and for the full LogART (OHS+LLR) are presented in Table 14.

Table 14: Comparison of performance (PPL) for different calibration dataset sizes under 3-bit per-channel weight quantization using LogART.

| Config. | Segments | OPT-125M | | | LLaMA2-7B | | |
|---|---|---|---|---|---|---|---|
| | | PPL (WT-2)* | PPL (C4) | Runtime | PPL (WT-2)* | PPL (C4) | Runtime |
| OHS | 32 | 34.29 | 32.17 | 17.0 s | 6.45 | 8.72 | 17.9 min |
| | 128 | 34.26 | 32.14 | 22.6 s | 6.42 | 8.67 | 20.6 min |
| OHS+LLR | 32 | 31.15 | 30.44 | 75.1 s | 6.14 | 8.55 | 1.24 hr |
| | 128 | 31.17 | 30.38 | 79.9 s | 6.09 | 8.52 | 1.30 hr |

* WT-2 refers to the WikiText-2 dataset.

# F  3-BIT PTQ PERFORMANCE OF LOGART ON VISION TRANSFORMERS

In the main body of the paper (Table 5), we present the 4-bit weight quantization performance of LogART on various vision transformer models. To further evaluate the robustness and effectiveness of our method under more aggressive compression settings, this section provides the corresponding results for 3-bit per-channel weight quantization. The experimental setup follows that of 4-bit experiments in the main text, with only the target bitwidth changed. The comparison results are presented in Table 15.

Table 15: Comparison of top-1 accuracy and GPU runtime (in minutes) for 3-bit and 4-bit per-channel weight quantization with LogART on vision transformers.

| W | ViT-Small | | ViT-Base | | DeiT-Tiny | | DeiT-Base | |
|---|---|---|---|---|---|---|---|---|
| | Acc(%) | Runtime | Accc(%) | Runtime | Accc(%) | Runtime | Accc(%) | Runtime |
| FP16 | 81.39 | - | 85.10 | - | 72.16 | - | 81.98 | - |
| 4-bit | 81.06 | 6.7 | 85.02 | 10.9 | 71.62 | 5.6 | 81.92 | 10.9 |
| 3-bit | 79.56 | 5.9 | 84.54 | 9.9 | 70.21 | 5.0 | 81.51 | 10.1 |

# G DETAILS OF LOGART AE

## G.1 LOGART AE DESIGN

While our proposed multi-base logarithmic quantizer enhances accuracy compared with fixed Log2 quantizer, it indeed introduces implementation challenges for the AE. Our LogART AE, shown in Figure 4, is customized to overcome these issues. It handles the computation involving $\sqrt{2}$ through the integrated HAF, and also performs efficient on-chip decoding of the dynamic multi-base values through a novel encoding and decoding scheme. The functionality is realized through a design consisting of four primary modules: decoder, signed arithmetic logic, approximate computing logic, and shifter.

The decoder and its associated encoding scheme efficiently manage the complexity introduced by the multi-base logarithmic quantizer. In a 4-bit example, each weight is encoded with a 1-bit sign and a 3-bit value code ($w_code$) that maps to one of the $n_1$ base-$\sqrt{2}$ larger-valued codes and $n_2$ smaller-valued base-2 codes. This is accompanied by 4-bit per-channel metadata that stores $n_2$ and a parity flag ($chk_even$). The $chk_even$ is decided by the parity of the maximum base-$\sqrt{2}$ exponent. During computation, the decoder uses simple combinational logic to process this information and output control signals for the multiplier-free AE: a 1-bit sign, 3 shift bits ($Shift_bits$), and a 1-bit approximation flag ($Approx_flag$). Table 16 provides concrete examples of this process.

Table 16: Encoding and decoding scheme examples.

| Weight | Base | After Scaling | | $w_code$ | | $Shift_bits$ | $Approx_flag$ |
|---|---|---|---|---|---|---|---|
| Example 1: $n_2$=4, $chk_even$=1, Scaling Factor=$2^{-8}$ | | | | | | | |
| $2^{-8}$ | 2 | $2^0$ | Encoding -> | 000 | Decoding-> | 0 | 0 |
| $2^{-7}$ | 2 | $2^1$ | | 001 | | 1 | 0 |
| $2^{-6}$ | 2 | $2^2$ | | 010 | | 2 | 0 |
| $2^{-5}$ | 2 | $2^3$ | | 011 | | 3 | 0 |
| $2^{-4.5}$ | $\sqrt{2}$ | $2^{3.5}$ | | 100 | | 3 | 1 |
| $2^{-4}$ | $\sqrt{2}$ | $2^4$ | | 101 | | 4 | 0 |
| $2^{-3.5}$ | $\sqrt{2}$ | $2^{4.5}$ | | 110 | | 4 | 1 |
| $2^{-3}$ | $\sqrt{2}$ | $2^5$ | | 111 | | 5 | 0 |
| Example 2: $n_2$=5, $chk_even$=0, Scaling Factor=$2^{-8}$ | | | | | | | |
| $2^{-8}$ | 2 | $2^0$ | Encoding -> | 000 | Decoding-> | 0 | 0 |
| $2^{-7}$ | 2 | $2^1$ | | 001 | | 1 | 0 |
| $2^{-6}$ | 2 | $2^2$ | | 010 | | 2 | 0 |
| $2^{-5.5}$ | $\sqrt{2}$ | $2^{2.5}$ | | 011 | | 2 | 1 |
| $2^{-5}$ | $\sqrt{2}$ | $2^3$ | | 100 | | 3 | 0 |
| $2^{-4.5}$ | $\sqrt{2}$ | $2^{3.5}$ | | 101 | | 3 | 1 |
| $2^{-4}$ | $\sqrt{2}$ | $2^4$ | | 110 | | 4 | 0 |
| $2^{-3.5}$ | $\sqrt{2}$ | $2^{4.5}$ | | 111 | | 4 | 1 |

The signed arithmetic logic operates based on the sign bit of the weight. For a positive weight, the activation is used directly. For a negative weight, the module computes the two's complement of the activation to perform the negation. This module is a crucial part of the arithmetic logic that ensures calculations involving negative weights are performed correctly and efficiently in hardware. The output of the signed arithmetic logic is then passed to the approximate computing logic.

The approximate computing logic and the shifter operate together to execute the final stage of the multiplier-free computation, supporting the multi-base feature of the LogART quantizer. The approximate computing logic is specifically designed to handle computations involving the $\sqrt{2}$ base. With $K = 2$ selected for our HAF due to its strong balance between hardware efficiency and accuracy, the multiplication by $\sqrt{2} \approx 1.5$ is implemented as $X + X/2$. The approximate computing logic takes an enabling signal, $Approx_flag$, from the decoder. If disabled, the activation passes through unchanged; otherwise, it adds the activation to half of its value. The shifter then performs the final multiplication-like step. It takes the output from the approximate computing logic and applies a left

bit-shift, with the number of shift positions determined by $Shift\_bits$ from the decoder. This single shift operation is equivalent to a multiplication by a power of 2, completing the computation and producing the final output value.

## G.2 LOGART AE BREAKDOWN

With all four modules working in concert, the LogART AE correctly computes the product for any weight in its multi-base system, thereby realizing a fully multiplier-free design. The overall area and power consumption for this complete AE are presented in Table 6. To provide a more granular analysis, Table 17 further breaks down the hardware cost into the key functional modules, as illustrated in the block diagram in Figure 4(e).

Table 17: Area and power breakdown of the LogART AE.

|  | LogART AE | Signed | Decoder | Approx(Add) | Shift |
|---|---|---|---|---|---|
| Area ($\mu$m$^2$) | 53.2 (100%) | 19.6% | 17.5% | 25.6% | 37.3% |
| Power ($\mu$W) | 3.45 (100%) | 20.7% | 21.1% | 20.0% | 38.2% |

The hardware breakdown reveals that the shifter is the dominant contributor to hardware cost, accounting for 37.3% of the total area and 38.2% of the power. This result is expected, as the variable shifter serves as the core computational unit in a LogART AE. The next largest contributors are the modules that enable our multi-base scheme: the approximate computing logic, which implements the hardware-friendly operation for the $\sqrt{2}$ base, is the second-largest by area (25.6%), while the decoder is the second-largest consumer of power (21.1%). Together, these two modules represent the necessary hardware investment to support the flexibility and high accuracy of our multi-base design. This breakdown validates our design choices, showing that even with this essential overhead, the LogART AE remains compact and power-efficient overall.

## H    THE USE OF LLMS

During the preparation of this manuscript, the authors utilized an AI language model to assist with improving the grammar, clarity, and overall readability of the text. The role of LLMs was strictly limited to that of a writing assistant for language polishing.

