# OpenReview forum: "LogART: Pushing the Limit of Efficient Logarithmic Post-Training Quantization"
_ICLR.cc/2026/Conference — ICLR 2026 Poster_

### Official Review · Reviewer_zR6Q · 2025-10-28

**Soundness:** 3
**Presentation:** 3
**Contribution:** 2
**Rating:** 6
**Confidence:** 3

**Summary:**

The paper introduces LogART, a novel post-training quantization (PTQ) method designed for logarithmic quantization of neural network weights. Unlike linear quantization, logarithmic quantization aligns better with weight distributions often found in large models, and enables hardware efficiencies through base-2 arithmetic. LogART innovates by combining learnable rounding in the logarithmic domain with an adaptive quantizer that supports outlier resilience, asymmetry, and multi-base logarithmic representations. The authors also propose a multi-level hyperparameter search and a hardware approximation function (HAF) to make their scheme efficient and hardware-friendly. They evaluate LogART on a variety of architectures, LLMs, CNNs, and ViTs, showing higher accuracy than prior log-based PTQ, and meaningful hardware-area and power benefits.

**Strengths:**

- LogART supports multi-base quantization, e.g., mixing base-2 and base-$\sqrt{2}$.


- The HAF (Hardware Approximation Function) approximates $\sqrt{2}$ with shift-add operations, trading off minimal accuracy loss for hardware efficiency. They simulate area and power in a 28 nm process, showing substantial savings compared to conventional designs.

- Experiments across LLMs, CNNs, and ViTs strengthen the claim of generality.

- The ablation studies clearly show the impact of each component (LLR, OHS, dynamic base, etc.).

**Weaknesses:**

- The evaluations focus primarily on 3-bit quantization. It would be useful to show how LogART performs at even lower (2-bit) weights and to compare it with other PTQ methods like QuIP[1].

- The paper mentions compatibility with activation quantization but does not deeply explore joint weight-activation quantization. Joint methods are often more relevant for real deployments.

[1] QuIP: 2-Bit Quantization of Large Language Models With Guarantees

**Questions:**

- Error Bound Analysis: do you have theoretical bounds on the error introduced by the HAF approximation of $\sqrt{2}$?

- Lower Bitwidth: have you experimented with 2-bit weight quantization? If so, how does LogART perform?

---

> ### Author Response · Authors · 2025-11-23
>
> We appreciate your positive feedback on our algorithmic and hardware contributions. In response to your specific concerns, we have: (1) **added 2-bit quantization experiments** that highlight LogART's superior performance; (2) **added joint weight-activation quantization experiments** to showcase compatibility; and (3) provided a rigorous three-level **error bound analysis for HAF**, including the gradient computation path, in Appendix C (*highlighted in green*). We hope these revisions fully address your questions and encourage you to favorably reconsider your score.
>
> **Weakness (W) and Question (Q)**
>
> **W1&Q2**: The evaluations focus primarily on 3-bit quantization. It would be useful to show how LogART performs at even lower (2-bit) weights and to compare it with other PTQ methods like QuIP. & Have you experimented with 2-bit weight quantization? If so, how does LogART perform?
>
> **A1**: As suggested, we have extended the LogART evaluation to include 2-bit weight quantization and a comparison with QuIP. LogART demonstrates significant improvements over QuIP in the 2-bit regime. The table below shows the PPL comparison of per-channel 2-bit weight quantization on LLMs using 128 random 2048-token calibration segments from C4.
>
> |        |                 | OPT-125M | LLaMA2-7B |
> |--------|-----------------|----------|-----------|
> | QuIP   | PPL(WikiText-2) | 347.4    | 27.13     |
> |        | PPL(C4)         | 177.4    | 31.33     |
> | LogART | PPL(WikiText-2) | 72.93    | 10.55     |
> |        | PPL(C4)         | 58.18    | 12.50     |
>
> These results highlight the superior representational capability of LogART's asymmetric, outlier-resilient logarithmic quantization grid, further enhanced by fine-grained element-wise learnable rounding optimization. This confirms LogART's effectiveness even at ultra-low-bit quantization.
>
> **W2**: The paper mentions compatibility with activation quantization but does not deeply explore joint weight-activation quantization. Joint methods are often more relevant for real deployments.
>
> **A2**: We sincerely appreciate the reviewer for highlighting this critical direction. As suggested, we added experiments integrating the LogART weight quantization with the activation quantization from AdaLog. We conducted joint quantization experiments using 4-bit weight and 4-bit activation (W4A4) under identical experimental settings. The table below compares against existing methods:
>
> | Method    | ViT-S | ViT-B | DeiT-S | DeiT-B |
> |-----------|-------|-------|--------|--------|
> | PTQ4ViT   | 42.57% | 30.69% | 34.08%  | 64.39%  |
> | APQ-ViT   | 47.95% | 41.41% | 43.55%  | 67.48%  |
> | RepQ-ViT  | 65.05% | 68.48% | 69.03%  | 75.61%  |
> | AdaLog    | 72.75% | 79.68% | 72.06%  | 78.03%  |
> | Ours      | 73.15% | 80.67% | 74.61%  | 79.28%  |
>
> As shown above, our method consistently achieves higher accuracy than other baselines in the W4A4 setting. While these results are encouraging, we believe the intricate dynamics of fully optimized joint quantization merit a dedicated investigation. Our work focuses on weight-only PTQ. Including joint quantization is beyond the scope of this paper and would overwhelm the current analysis. We plan to explore joint quantization in future work.

---

> ### Author Response · Authors · 2025-11-23
>
> **Q1**: Error Bound Analysis: do you have theoretical bounds on the error introduced by the HAF approximation of?
>
> **A1**: We thank the reviewer for this valuable question. In the revised Appendix C, we have added a rigorous theoretical analysis of the HAF-induced error bound. Our analysis unfolds in four steps:
> 1. **$\\sqrt{2}$ Approximate Error**: First, we prove that the greedy iterative approach of the Signed Dyadic Expansion (SDE) approximation guarantees that the residual error $\epsilon_K = \left| r_{K+1} \right|$ is strictly bounded by the precision of the $K$-th term:
> $$\epsilon_K < \frac{1}{2^{d_K}}.$$
> 2. **Weight Error Bound**: Next, use the  $\sqrt{2}$ approximation to compute the disturbed quantized weight $\Delta \widehat{\mathbf{W}}$. Applying the $\sqrt{2}$ error bound yields the theoretical upper bound of the weight error:
> $$\left| \Delta \widehat{\mathbf{W}} \right| < \left| \mathbf{\widehat{W}} \right| \odot \frac{1}{\sqrt{2} \cdot 2^{d_K}} \odot \mathbf{M}.$$
> 3. **Reconstruction Error Bound**: We further propagate weight error to the task-aware reconstruction loss $\mathcal{L}$. The theoretical upper bound for the HAF-induced error in the reconstruction loss diminishes exponentially with respect to $d_K$:
> $$ \mathbb{E} \left[ \mathcal{L} \left( \mathbf{\Delta \widehat{\mathbf{W}}} \right) \right] < \frac{1}{2^{1 + 2d_K}} \cdot \text{tr} \left( ( \widehat{\mathbf{W}} \odot \mathbf{M} ) \cdot \mathbf{H} \cdot (\widehat{\mathbf{W}} \odot \mathbf{M})^\top \right). $$
> 4. **Compensation by integrating HAF into LLR**: While a larger $K$ reduces reconstruction loss and potentially improves accuracy, it incurs increasing hardware overhead. To resolve this trade-off, we integrate the HAF-induced error directly into the LLR learning process, allowing the model to compensate via element-wise optimized rounding decisions. The deterministic HAF error factor is explicitly included in the gradient computation, allowing the LLR optimization to theoretically learn to compensate for it:
> $$ \frac{\partial \widehat{\mathbf{W}}'}{\partial \widehat{\mathbf{W}}} = \mathbf{1} + (\gamma - 1) \mathbf{M}.$$
>
> Ultimately, we prove that the **reconstruction error caused by HAF is theoretically bounded** and successfully **absorbed during the LLR process**.

---

### Official Review · Reviewer_HfTW · 2025-10-29

**Soundness:** 3
**Presentation:** 3
**Contribution:** 3
**Rating:** 6
**Confidence:** 4

**Summary:**

This paper proposes a novel adaptive weight rounding technique called LogART for improving logarithmic post-training quantization (logarithmic PTQ) performance on large language models (LLMs). The method introduces key components including dynamic base selection, scaling factor selection, adaptive base selection, and learnable weight rounding, significantly enhancing the performance of 3-bit channel-wise weight quantization models while maintaining hardware efficiency.

**Strengths:**

(1) First proposes a learnable weight rounding technique specifically for logarithmic PTQ, solving the problem that existing methods cannot apply learnable weight rounding in non-linear quantization.
(2) Designs multiple combinable components (DBS, SFS, ABS, LLR), with experiments showing that these components produce synergistic effects when used together, significantly reducing perplexity (PPL).
(3) Extensively validates LogART's effectiveness on LLMs such as OPT-125M and LLaMA2-7B, demonstrating significant performance improvements under 3-bit quantization.
(4) Analyzes the impact of calibration datasets on performance, proving LogART's robustness to calibration data sources.

**Weaknesses:**

(1) The paper lacks theoretical analysis of the proposed method, particularly in-depth explanation of interactions between components.
(2) Experiments are primarily focused on LLMs, with insufficient validation on other model types such as CNNs or ViTs, limiting the method's generalizability.
(3) The paper does not discuss the additional computational overhead of LogART in practical deployment, especially the extra training steps required for learning rounding parameters, which may contradict the core goal of quantization techniques to simplify models and improve hardware efficiency.

**Questions:**

(1) The paper mentions that LogART's components can be used in combination, but does not detail how these components work together, nor whether there exists an optimal combination.
(2) The paper uses WikiText-2 and C4 as calibration datasets in experiments, but does not deeply explore the impact mechanism of different calibration datasets on performance, nor how to choose the best calibration dataset.

---

> ### Author Response · Authors · 2025-11-21
>
> Thank you for your insightful comments. Based on your suggestions, we have made the following updates: (1) included a theoretical analysis of how LogART components work together; (2) conducted additional experiments on CNNs and ViTs, demonstrating LogART's strong generalizability; (3) clarified the hardware efficiency benefits of LogART; and (4) added a discussion on the choice of calibration dataset. We have incorporated these results into the revision (*highlighted in blue*) to strengthen the work, and we hope you will consider raising your score in light of these improvements. Please let us know if you have any follow-up questions.
>
> **Weakness (W)**
>
> **W1**: The paper lacks theoretical analysis of the proposed method, particularly in-depth explanation of interactions between components.
>
> **A1**: We sincerely appreciate the reviewer for this inspiring comment, which motivated us to significantly strengthen the theoretical foundations of our work. In the revised manuscript, we have added a new section, Appendix B.3: Theoretical Analysis of Component Synergy, to provide a rigorous mathematical derivation of the interactions between the Optimized Hyperparameter Search (OHS) and Learnable Logarithmic Rounding (LLR). The reconstruction error can be expressed as a Frobenius norm by incorporating the matrix square root of the Hessian $\mathbf{H} = \mathbb{E}[\mathbf{X}\mathbf{X}^\top]$ directly into the norm:
> $$\mathcal{L} = \text{tr}\left( \Delta \mathbf{W} \mathbf{H} \Delta \mathbf{W}^\top \right) = \left\\| \Delta \mathbf{W} \mathbf{H}^{\frac{1}{2}} \right\\|_F^2.$$
>
> The **triangle inequality** establishes an upper limit on the quantization error norm, separating it into two distinct geometric terms:
> $$ \left\\| \Delta \mathbf{W} \mathbf{H}^{\frac{1}{2}} \right\\|\_F^2 \le \Big( \underbrace{\left\\| (\mathbf{W} - \Pi\_{\mathcal{C}(\theta)}^{\mathbf{H}}(\mathbf{W})) \mathbf{H}^{\frac{1}{2}} \right\\|\_F}\_{\mathcal{E}\_1 \text{ (OHS)}} + \underbrace{\left\\| (\Pi\_{\mathcal{C}(\theta)}^{\mathbf{H}}(\mathbf{W}) - \mathbf{\widetilde{W}}) \mathbf{H}^{\frac{1}{2}} \right\\|\_F}_{\mathcal{E}\_2 \text{ (LLR)}} \Big)^2 $$
>
> Our analysis reveals that OHS and LLR target these decoupled error terms, $\mathcal{E}\_1$ and $\mathcal{E}\_2$, respectively. Within OHS, **SFS aligns the global scale** through $s_{\rm of}$, **DBS optimizes local density** using $n_1$, and **ABS adjusts support bounds** via $l_a$. These three components work synergistically to minimize the intrinsic discretization error $\mathcal{E}_1$. Crucially, we demonstrate that OHS searches for the optimal quantization grid that minimizes $\mathcal{E}_1$, while LLR learns the optimal rounding decision to minimize the residual error $\mathcal{E}_2$ on that established grid. This formalizes the **synergistic interaction between the components**.
>
> **W2**: Experiments are primarily focused on LLMs, with insufficient validation on other model types such as CNNs or ViTs, limiting the method's generalizability.
>
> **A2**: We appreciate the reviewer raising the importance of generalizability. We would like to respectfully clarify that **generalizability is one of the strengths of LogART**. We have also expanded our experimental evaluation on CNNs and vision transformers in the revision.
>
> Extensive experiments have confirmed the superiority of LogART on CNNs and vision transformers:
> 1. Performance of each component of LogART: The newly added **ablation study experiments** in Tables 2 and 12 confirms that our key components are effective across CNN and vision transformer architectures, particularly in recovering performance for sensitive compact models like MobileNetV2.
> 2. Superior accuracy and efficiency compared to SOTA: As shown in Tables 4 and 5, LogART is **compared with many state-of-the-art PTQ methods for CNN and vision transformers**, such as BRECQ and FlexRound for CNNs, and APHQ and AdaLog for vision transformers. LogART consistently outperforms SOTA methods in terms of top-1 accuracy on ImageNet. Compared with advanced PTQ methods with learnable rounding, LogART achieves a speedup of over 3.9$\times$.
> 3. Win-win of accuracy and hardware efficiency: As shown in Tables 6, 9, and 10, the **experiments on $K$ in HAF and the arithmetic element (AE)** confirm that the proposed HAF compensates for the error introduced by the hardware approximation of $\sqrt{2}$. Therefore, the LogART AE can achieve superior hardware efficiency while maintaining accuracy on CNNs and vision transformers.
>
> These results strongly validate that LogART is a generalized framework capable of handling diverse architectures, not limited to LLMs.

---

> ### Author Response · Authors · 2025-11-21
>
> **W3**: The paper does not discuss the additional computational overhead of LogART in practical deployment, especially the extra training steps required for learning rounding parameters, which may contradict the core goal of quantization techniques to simplify models and improve hardware efficiency.
>
> **A3**: We thank the reviewer for this critical comment regarding computational overhead. We clarify that **the computational overhead in practical deployment has already been discussed**. LogART achieves both low one-off offline quantized model production cost and reduced recurring inference hardware cost.
>
> It is important to clarify the distinction:
>  - **One-off offline quantized model production cost** (evaluated through **GPU runtime**): The mentioned extra training steps are a one-off cost incurred only during the model production phase, not during practical inference after deployment. We clarify that the GPU runtime reported in Tables 1-5, 7-12, 14, and 15 is exactly the model production time and **explicitly includes these training steps**. LogART is highly efficient, reducing GPU runtime by $>$2.2$\times$ on LLMs. For CNNs and vision transformers, compared with SOTA logarithmic PTQ methods without rounding learning, LogART achieves up to a 40.42% accuracy improvement at the cost of only 0.8–4.7 minutes.
>  - **Recurring inference hardware cost** (evaluated through **AE hardware cost**): During practical usage, the learned rounding parameters are frozen into the model weights. The inference hardware cost is determined by the AE. As demonstrated in our hardware analysis (Table 6), the LogART AE achieves significant hardware savings, with a 43.23% reduction in area and 61.16% reduction in power.
>
> Viewed from a long-term perspective, the extra training time is a one-off setup cost that is **amortized over millions of inference runs**, resulting in a negligible marginal cost. LogART achieves a competitive balance, offering both fast offline quantized model production and superior online inference hardware efficiency.
>
> **Question (Q)**
>
> **Q1**: The paper mentions that LogART's components can be used in combination, but does not detail how these components work together, nor whether there exists an optimal combination.
>
> **A1**: We appreciate the reviewer for raising this important point regarding component combination. As detailed in our answer A1 to W1, we have included a theoretical analysis of component synergy in Appendix B.3.
>
> We decompose the quantization error into two decoupled terms: intrinsic discretization error $\mathcal{E}\_1$ and rounding-induced error $\mathcal{E}\_2$. DBS, SFS, and ABS, within OHS work synergistically to minimize $\mathcal{E}\_1$ by searching for the optimal quantization grid. Once the grid is established, LLR learns the rounding decision to minimize $\mathcal{E}\_2$.
>
> This confirms that the **full integration of all proposed components** minimizes the upper limit of the reconstruction error, making it the **theoretically optimal configuration**. Empirically, the ablation studies in Tables 11 and 12 validate this conclusion, demonstrating the significant performance advantage achieved by combining all components.
>
> **Q2**: The paper uses WikiText-2 and C4 as calibration datasets in experiments, but does not deeply explore the impact mechanism of different calibration datasets on performance, nor how to choose the best calibration dataset.
>
> **A2**: We have added a sensitivity analysis of the calibration data source in Appendix E.1 to investigate the impact mechanism and guide dataset selection.
>
> As shown in Table 13, we observe a distinct **in-domain alignment pattern**. Models calibrated on WikiText-2 generally achieve slightly lower PPL on the WikiText-2 test set (with minor exceptions on OPT-1.3B and OPT-6.7B). Models calibrated on C4 consistently perform better on the C4 test set. This indicates that the calibration set helps the quantization parameters adapt to the specific linguistic distribution of the target domain.
>
> The performance difference stems from domain alignment and outlier coverage. **C4 provides high linguistic diversity**, covering a wider range of activation outliers, which ensures better generalization. In contrast, **WikiText-2 offers a highly consistent, formal distribution**. When used for both calibration and evaluation, WikiText-2 minimizes the distribution shift, leading to slightly lower PPL due to in-domain overfitting.
>
> Crucially, despite these variations, LogART exhibits strong robustness. The performance variance across different calibration sources is marginal (typically <1.0 PPL), and the model maintains high accuracy regardless of the source. Regarding **calibration dataset selection strategy**, for general-purpose deployment and fair SOTA comparison, we recommend using a large-scale, diverse dataset like C4. However, for tasks targeting a specific domain, utilizing in-domain calibration data yields the best fine-grained performance.

---

### Official Review · Reviewer_KiDj · 2025-10-31

**Soundness:** 3
**Presentation:** 3
**Contribution:** 2
**Rating:** 4
**Confidence:** 3

**Summary:**

The paper presents several methods that combined together improve the current SoTA on logarithm PTQ:
- Learnable Logarithmc Rounding (LLR): train the best rounding on calibration data set;
- Dynamic Base Quantizer (DBS): combine sqrt(2) and 2 bases;
- Asymmetric Quantizer (ABS): induce an asymmetry by clamping close to 0 values;
- Scaling Factor (SFS): train a scaling factor for clamping on a calibration data set, to better handle outliers.

**Strengths:**

- The paper advances the SotA on logarithmic quantization, mostly by combining/transfering known methods;
- Adequate systematic ablation study is made (available in the supplementary material, albeit only for Transformers);
- The paper evaluates its methods on both CNN and Transformers models;
- Although secondary in the main paper, a study of hardware implementation of the dual sqrt(2) and 2 base log is performed.

**Weaknesses:**

- I did not understood the asymmetric quantizer scheme: it seems to me that your are only clamping small values after quantization, therefore actually reducing the effective number of bits! I don't understand how this can improve things and indeed, from your ablation studies, it seems to degrade the PPL when not used in combination with LLR.
- The paper is very incremental and essentially combines/transfers known methods from the SoTA.

**Questions:**

If you could explain how and why ABS work.
Also, you only made full ablation study on Transformers. I would be very interested to see a similar full ablation study on CNN.

---

> ### Author Response · Authors · 2025-11-21
>
> Thank you for your valuable feedback. We have integrated your suggestions into the revision (*highlighted in orange*), which further enhances the paper quality. Below is a detailed response to each question.
>
> **Weakness (W) and Questions (Q)**
>
> **W1&Q1**: I did not understood the asymmetric quantizer scheme: it seems to me that your are only clamping small values after quantization, therefore actually reducing the effective number of bits! I don't understand how this can improve things and indeed, from your ablation studies, it seems to degrade the PPL when not used in combination with LLR. & If you could explain how and why ABS work.
>
> **A1**: We appreciate the opportunity to clarify the mechanism of Asymmetric Bound Search (ABS). We clarify that ABS does not reduce the effective number of bits nor simply clamp small values. Instead, **ABS acts as a dynamic range shifter in the logarithmic domain that maximizes the utilization of the available bit-width**. To help understanding, we have revised the description of the asymmetric quantizer and added Figure 3 in Section 3.2.
>
> First, we would like to clarify the challenges. As shown in Figure 3(c), an asymmetric quantizer in linear PTQ can be easily achieved through a simple zero-point that shifts the upper and lower clamping bounds. However, due to the absolute-value operator that appears at the very beginning of the logarithmic quantization in Eq. (1), (3), (5), and (12), **existing logarithmic PTQ methods are inherently symmetric**. As a result, they fail to capture naturally asymmetric weight distributions. Furthermore, **logarithmic quantization shows non-uniform spacing near zero**, making the simple bound-shifting strategy used in linear quantization inapplicable in the logarithmic domain.
>
> To overcome these challenges, ABS is introduced as the **first** asymmetric logarithmic quantizer that allocates different numbers of codes to positive and negative values. As shown in Figure 3(e), ABS accomplishes this by shifting the lower bound, thereby **reallocating unused fixed-points to finer-grained fixed-points near zero**. Importantly, ABS redistributes the available discrete codes from regions with no data to regions where the weights are concentrated.
>
> Regarding the concern about performance degradation, we respectfully direct the reviewer to the full ablation study in Table 11 and 12 of Appendix D. A pairwise comparison confirms its consistent benefit. Comparing Row 1 vs. 3, 2 vs. 4, 5 vs. 7, 6 vs. 8, 9 vs. 11, 10 vs. 12, and 13 vs. 15, 14 vs. 16 shows that, when all other components are held constant, **enabling ABS always results in lower PPL (lower the better)**.
>
> **W2**: The paper is very incremental and essentially combines/transfers known methods from the SoTA.
>
> **A2**: We argue that our work addresses unsolved challenges in logarithmic PTQ through the proposed innovations rather than through an incremental combination of existing ideas. Many **existing techniques cannot be applied to logarithmic PTQ without the specific innovations we introduce**:
>
> 1. **Inherent symmetric quantization grids** as discussed in A1. To overcome these challenges, we propose the **first asymmetric logarithmic quantizer** through ABS.
> 2. **Reliance on RTN**. Key difficulties include the non-linearity of the logarithmic mapping, the non-differentiability of rounding in the logarithmic domain, and the discrete nature of dynamic bases. LogART is the **first to address the non-learnability of logarithmic rounding and the non-differentiability in a multi-base dynamic setting** through the proposed Dynamic Base Search (DBS) and Learnable Logarithmic Rounding (LLR).
> 3. **Hardware deployment challenge**. LogART designs a customized multiplier-free Arithmetic Element (AE) for shift–add-based inference. LogART also propose the Hardware Approximation Function (HAF) module, enabling the **rounding learning process to absorb hardware-induced errors for the first time**. Together, this co-design achieves a win-win outcome, delivering both high accuracy and hardware efficiency.
> 4. **Vast search space and optimization complexity**. Scaling Factor Search (SFS), DBS, ABS and LLR together create a massive high-dimensional search space. we propose a **multi-level Optimized Hyperparameter Search (OHS) and studies its synergetic effect with LLR**. Hierarchically establishing the optimal grid, OHS effectively prunes the search space. This acts as an enabler for LLR to focus on fine-grained optimization, resulting in accelerated convergence and enhanced optimization performance.
>
> Therefore, the LogART components are not ‘transferred’ but are tailored innovations for logarithmic PTQ. The significance of LogART is further supported by empirical evidence:
>  - LogART consistently delivers superior accuracy across LLMs, ViTs, and CNNs, while achieving faster convergence.
>  - LogART achieves over a 40% reduction in AE area and power consumption, validating its practical hardware impact.

---

> ### Author Response · Authors · 2025-11-21
>
> **Q2**: Also, you only made full ablation study on Transformers. I would be very interested to see a similar full ablation study on CNN.
>
> **A2**: Thanks for the suggestion, we fully agree that a full ablation study on CNN could enhance the generalizability of our work. We have added full ablation experiments and analysis for CNNs, highlighted in revised Appendix D, to further strengthen LogART’s versatility.
>
> |     |     |     |     | ResNet18 |             | MobileNetV2 |             |
> |-----|-----|-----|-----|----------|-------------|-------------|-------------|
> | DBS | SFS | ABS | LLR | Top-1 Acc(%)   | Runtime (s) | Top-1 Acc(%)      | Runtime (s) |
> | $\\times$   | $\\times$   | $\\times$   | $\\times$   | 31.53    | 0.9         | 1.22        | 1.9         |
> | $\\times$   | $\\times$   | $\\times$   | $\\checkmark$   | 69.75    | 60.8        | 68.58       | 133.0       |
> | $\\times$   | $\\times$   | $\\checkmark$   | $\\times$   | 32.04    | 1.0         | 1.25        | 1.9         |
> | $\\times$   | $\\times$   | $\\checkmark$   | $\\checkmark$   | 69.78    | 61.0        | 68.62       | 133.4       |
> | $\\times$   | $\\checkmark$   | $\\times$   | $\\times$   | 67.13    | 2.8         | 58.43       | 5.6         |
> | $\\times$   | $\\checkmark$   | $\\times$   | $\\checkmark$   | 69.82    | 62.7        | 68.96       | 138.8       |
> | $\\times$   | $\\checkmark$   | $\\checkmark$   | $\\times$   | 67.29    | 2.8         | 58.48       | 5.6         |
> | $\\times$   | $\\checkmark$   | $\\checkmark$   | $\\checkmark$   | 69.89    | 62.8        | 69.02       | 138.9       |
> | $\\checkmark$   | $\\times$   | $\\times$   | $\\times$   | 68.45    | 1.0         | 66.91       | 2.3         |
> | $\\checkmark$   | $\\times$   | $\\times$   | $\\checkmark$   | 70.50    | 61.7        | 71.24       | 134.4       |
> | $\\checkmark$   | $\\times$   | $\\checkmark$   | $\\times$   | 69.04    | 1.0         | 67.00       | 2.3         |
> | $\\checkmark$   | $\\times$   | $\\checkmark$   | $\\checkmark$   | 70.59    | 61.5        | 71.25       | 134.6       |
> | $\\checkmark$   | $\\checkmark$   | $\\times$   | $\\times$   | 69.69    | 10.8        | 69.47       | 22.0        |
> | $\\checkmark$   | $\\checkmark$   | $\\times$   | $\\checkmark$   | 70.69    | 71.1        | 71.55       | 153.4       |
> | $\\checkmark$   | $\\checkmark$   | $\\checkmark$   | $\\times$   | 69.89    | 11.6        | 69.86       | 24.4        |
> | $\\checkmark$   | $\\checkmark$   | $\\checkmark$   | $\\checkmark$   | 70.79    | 72.3        | 71.62       | 156.6       |
>
> Similar to the case of LLMs, the full ablation results on CNNs clearly illustrate the synergistic and cumulative effects of the LogART components. **While each module provides a benefit in isolation, they are most effective when combined**. Regardless of the starting component or any combination of components, adding an additional module always leads to further improvement in accuracy.

---

### Author Response · Authors · 2025-11-27
**General Response: revision update notice**

Dear Reviewers,

We sincerely thank the reviewers for their insightful and constructive feedback.

We are encouraged to see that all reviewers acknowledged our contribution in pushing the limits of logarithmic PTQ in efficiency, robustness, and performance. Aligned with the reviewers’ comments, our work addresses non-learnable rounding and non-differentiable multi-base configuration through Learnable Logarithmic Rounding (LLR) and Dynamic Base Search (DBS) [Reviewers **KiDj**, **HfTW**, **zR6Q**], resolves inherent symmetry using the proposed asymmetric logarithmic quantizer and Asymmetric Bound Search (ABS) [Reviewers **HfTW**, **zR6Q**], handles outliers via Scaling Factor Search (SFS) [Reviewers **KiDj**, **HfTW**, **zR6Q**], and enhances practical hardware benefits enabled by the Hardware Approximation Function (HAF) [Reviewers **KiDj**, **zR6Q**]. Ablation studies showing component synergy [Reviewers **KiDj**, **HfTW**], analysis of calibration dataset impact [Reviewer **HfTW**], and extensive evaluations on LLMs, CNNs, and ViTs [Reviewers **KiDj**, **zR6Q**] further confirm the generality and robustness of our method.

We greatly appreciate the reviewers’ efforts in evaluating our work, and their suggestions are invaluable for improving the paper. These include: adding ablation studies on more architectures [Reviewers **KiDj**, **HfTW**], incorporating new experiments on 2-bit quantization and joint weight-activation quantization [Reviewer **zR6Q**]; providing theoretical analysis of component interactions [Reviewer **HfTW**], HAF-induced error bounds [Reviewer **zR6Q**], and calibration dataset selection [Reviewer **HfTW**]; and clarifying the computational overhead in practical deployment [Reviewer **HfTW**]. All these recommendations have been integrated into the revised manuscript and highlighted.

Below, we provide detailed point-by-point responses to each of the reviewers' comments. If you still have concerns about specific aspects, please feel free to discuss them with us at any time.

Sincerely,
The Authors

---

### Author Response · Authors · 2025-12-04
**Final Remarks by the Authors**

Dear ACs, SACs, and PCs,

Thank you for your dedication and the considerable effort involved in ensuring the success of ICLR. We write to provide a concise summary of our paper’s status following the rebuttal phase.

We are encouraged by the positive feedback and the reviewers' recognition of our work's originality, generalizability, and applicability. In particular, **Reviewer HfTW** remarked that our work proposes a “novel” solution to the problem that “existing methods cannot apply in nonlinear quantization,” and “extensively validates the effectiveness” through “significant performance improvements”. **Reviewer KiDj** noted that our work “advances the SotA on logarithmic quantization.” **Reviewer zR6Q** highlighted that our work “innovates” in the logarithmic domain, offering both “higher accuracy than prior log-based PTQ” and “meaningful hardware benefits.”

Although the interactive discussion phase is no longer possible, we believe our rebuttal has fully addressed the raised concerns and that the reviews have significantly strengthened the paper. Below, we summarize the changes and clarifications made to the manuscript:

1. **Clarification of misunderstandings**

   Reviewer KiDj’s concern regarding 'transferring known methods' stems from a fundamental misconception: the reviewer overlooked that similar techniques studied in linear PTQ are mathematically incompatible with the logarithmic domain. Therefore, a simple 'combination' or 'transfer' is impossible. Our work introduces specific innovations to address these incompatibilities (a contribution explicitly recognized by Reviewer HfTW and Reviewer zR6Q), and tailors novel mechanisms, such as the dynamic base and hardware-aware function, to achieve both accuracy and practical hardware merit. Furthermore, we corrected the reviewer's confusion in ABS between fixed-point reallocation and simple clamping, and addressed the misreading of the PPL results.

   Reviewer HfTW’s concern regarding computational overhead largely arises from overlooking the GPU runtime and AE hardware cost data, which were included in the original submission. We clarified that the reported GPU runtime measures the one-off offline production cost, while the AE cost measures the recurring inference overhead. LogART achieves both low one-off offline quantized model production cost and reduced recurring inference hardware cost.

2. **Expanded theoretical analysis**

   Component Synergy (Reviewer HfTW): A theoretical analysis using the triangle inequality to establish an upper bound on the quantization error norm, mathematically confirming the synergistic interaction between components.

	HAF Error Bounds (Reviewer zR6Q): A rigorous four-step error bound analysis covering the $\sqrt{2}$ approximation error, weight error bound, reconstruction error bound, and the compensation mechanism via integrating HAF into LLR.

	Calibration Dataset Analysis (Reviewer HfTW): We provided an analysis of the calibration dataset's impact mechanisms (specifically domain alignment and outlier coverage) and offered guidance on dataset selection.

3. **Expanded experiments**

   We broadened our evaluation to include: ablation studies on a wider range of architectures (Reviewer KiDj, HfTW), 2-bit weight quantization experiments (Reviewer zR6Q), joint weight-activation quantization experiments (Reviewer zR6Q).

   These results confirm our conclusion: our proposed components exhibit clear synergy, and through the multi-level search strategy and optimization learning, LogART pushes the limits of logarithmic PTQ in accuracy, efficiency, generalizability, and hardware-friendliness.

Having addressed all major concerns, we sincerely thank again the reviewers for their constructive comments in refining this work. We also deeply appreciate your dedication to ensure the quality of ICLR 2026.

Sincerely,

The Authors

---

### Meta-Review · Area_Chair_GRwm · 2026-01-01

**Summary:**

This paper proposes **LogART**, a post-training quantization framework that makes **logarithmic PTQ** both more accurate and hardware-friendly by introducing **learnable rounding in the logarithmic domain** and a **distribution-aware search** over improved log grids.

Through extensive experiments on **LLMs, CNNs, and ViTs**, the authors show that combining **DBS/SFS/ABS** (grid construction: dynamic multi-base, outlier handling, and asymmetry) with **LLR** (task-aware learnable log rounding) yields consistent gains, and that a **hardware approximation function (HAF)** enables multiplier-free deployment while bounding/compensating approximation error.

I recommend **Accept (poster)**: the rebuttal addresses the main concerns (ABS misunderstanding, “transfer” misconception, theory of component synergy, overhead clarification, added ablations, and added 2-bit + joint W/A results), and the work provides a meaningful step forward for practical **hardware-efficient logarithmic quantization**.

**Reviewer Concerns:**

* **“Incremental / transferring known methods” (KiDj)**: **Addressed by the rebuttal.** The authors argue the “transfer” framing stems from a misconception: key linear-PTQ ideas are *mathematically incompatible* with the logarithmic domain due to nonlinearity, rounding non-differentiability, and discrete/multi-base structure. They emphasize that LogART introduces *domain-specific* mechanisms (LLR + DBS + hardware-aware design) that enable learnable rounding and multi-base configuration in log-PTQ rather than merely combining existing tricks.

* **ABS misunderstanding / “just clamping reduces effective bits” (KiDj)**: **Addressed by the rebuttal.** The authors clarify ABS is **not** post-quantization clamping that reduces effective precision; instead it is a **log-domain dynamic range/code reallocation** mechanism that enables *asymmetric* code allocation (positive vs negative) and reallocates unused fixed-points toward regions with higher weight density (especially near zero). They also point to pairwise ablations showing ABS consistently improves PPL when controlling for other components.

* **Missing full ablation beyond Transformers (KiDj, HfTW)**: **Addressed by the rebuttal.** The authors state they added **full ablation on CNNs** (e.g., ResNet18 / MobileNetV2) and expanded ablations across a wider range of architectures to support generality and component synergy beyond LLMs/Transformers.

* **Lack of theoretical explanation of component synergy (HfTW)**: **Addressed by the rebuttal.** The authors added a **triangle-inequality-based analysis** that upper-bounds quantization error and formalizes the synergy: OHS (SFS/DBS/ABS) minimizes the intrinsic discretization/grid error while LLR minimizes the residual rounding-induced error on the chosen grid.

* **Calibration dataset impact and selection guidance (HfTW)**: **Addressed by the rebuttal.** The authors added an analysis explaining *why* calibration data matters (domain alignment + outlier coverage), reported sensitivity results, and provided practical guidance (diverse corpora for general deployment vs in-domain calibration for best target-domain PPL).

* **Computational overhead / practicality (HfTW)**: **Addressed by the rebuttal (with a minor camera-ready clarity opportunity).** The authors clarify the key distinction:
  - **GPU runtime** is a **one-off offline** model production cost (including the extra learning steps), and
  - **AE cost** reflects **recurring inference** overhead.
  They argue LogART achieves both low offline production cost and lower inference hardware cost, and that the one-off cost is amortized. A small remaining improvement is to make this offline-vs-inference distinction extremely explicit and prominent in the camera-ready narrative.

* **HAF approximation error bounds (zR6Q)**: **Addressed by the rebuttal.** The authors report adding a rigorous multi-step bound covering: (i) approximation error, (ii) induced weight error, (iii) reconstruction error propagation, and (iv) how integrating HAF into LLR allows compensation during optimization (including the gradient path).

* **Lower bit-width (2-bit) and comparison to QuIP (zR6Q)**: **Addressed by the rebuttal.** The authors state they added **2-bit weight quantization** experiments and a **QuIP comparison**, showing substantial PPL gains for LogART in the 2-bit regime.

* **Joint weight–activation quantization (zR6Q)**: **Addressed by the rebuttal (scope note remains).** The authors state they added **W/A joint quantization experiments** (e.g., W4A4 integrating with AdaLog activations) to demonstrate compatibility. They also note that fully optimized joint quantization is beyond the paper’s main scope (weight-only PTQ), which remains a reasonable scope boundary rather than an unresolved weakness.

**Reviewer Scores:**

## Reviewer KiDj: 4 → 6
Why: Their key objections were (i) “incremental/transfer,” and (ii) misunderstanding ABS. The authors’ explanation and the ABS formulation/visualization (Figure 3) directly address the misconception (ABS reallocates codes asymmetrically rather than “just clamping”), and the argument that linear PTQ techniques are not directly transferable to the logarithmic domain is consistent with the paper’s framing.

## Reviewer HfTW: 6 → 6
Why: This reviewer is already supportive and the remaining concerns (theory, broader validation, and overhead clarity) are largely *additive* rather than blocking. Given their current “marginally above threshold” stance, they would most likely keep the same score: the added theoretical discussion, extra CNN/ViT experiments, and clearer separation of one-off calibration/runtime vs inference cost strengthen confidence, but probably not enough to justify a full-point increase.

## Reviewer zR6Q: 6 → 6
Why: The review is already positive and primarily requests additional evidence at lower bit-widths (2-bit), joint weight–activation quantization, and sharper HAF error-bound discussion. Even if these additions are incorporated cleanly, the reviewer’s current position is already “accept-leaning,” so the most likely outcome is that they maintain the same score rather than increase it, especially since the core contribution rating is already conservative and the paper’s main scope remains weight-only logarithmic PTQ.

---

### Decision · Program_Chairs · 2026-01-26

Accept (Poster)